# Physical–Chemical and Metataxonomic Characterization of the Microbial Communities Present during the Fermentation of Three Varieties of Coffee from Colombia and Their Sensory Qualities

Laura Holguín-Sterling [1], Bertilda Pedraza-Claros [1,2], Rosangela Pérez-Salinas [2], Aristófeles Ortiz [3], Lucio Navarro-Escalante [4,†] and Carmenza E. Góngora [4,*]

1 Department of Post Harvesting, National Coffee Research Center, Cenicafé, Manizales 170009, Colombia; lhsterling22@gmail.com (L.H.-S.); ber.pedraza@mail.udes.edu.co (B.P.-C.)
2 Cienciaudes Research Group, Faculty of Health Science, Universidad de Santander, Valledupar 200004, Colombia; ros.perez@mail.udes.edu.co
3 Department of Plant Physiology, National Coffee Research Center, Cenicafé, Manizales 170009, Colombia; aristofeles.ortiz@cafedecolombia.com
4 Department of Entomology, National Coffee Research Center, Cenicafé, Manizales 170009, Colombia; lucio.navarro@austin.utexas.edu
* Correspondence: carmenza.gongora@cafedecolombia.com; Tel.: +57-3154869466
† Current address: Department of Molecular Biosciences, University of Texas at Austin, Austin, TX 78712, USA.

**Abstract:** The microbial composition and physical-chemical characteristics were studied during the coffee fermentation of three *Coffea arabica* L. varieties, Var. Tabi, Var. Castillo General® and Var. Colombia. Mucilage and washed coffee seeds samples were collected at different stages of fermentation. Mucilage microbiology characterization and metataxonomic analysis were performed using 16S rDNA sequencing to determine bacterial diversity and ITS sequencing for fungal diversity. Additionally, the microorganisms were isolated into pure cultures. The molecular diversity analyses showed similarities in microorganisms present during the fermentation of Var. Castillo General and Var. Colombia, which are genetically closely related; mixed-acid bacteria (Enterobacteriaceae, *Tatumella* sp.) and lactic acid bacteria (*Leuconostoc* sp., *Weissella* sp. and *Lactobacillaceae*) were common and predominant, while in Var. Tabi, acetic acid bacteria (*Gluconobacter* sp. and *Acetobacter* sp.) and *Leuconostoc* sp. were predominant. At the end of the fermentation period, the fungi *Saccharomycodaceae*, *Pichia* and *Wickerhamomyces* were found in Var. Castillo General and Var. Colombia, while in Var. Tabi, *Saccharomycodaceae*, *Pichia* and *Candida* were recorded. Sensory analyses of the coffee beverages were carried out (SCA methodology) for all samples. Var. Tabi had the highest SCA score, between 82.7 and 83.2, while for Var. Colombia, the score ranged between 82.1 and 82.5. These three coffee varieties showed potential for the production of specialty coffees influenced by spontaneous fermentation processes that depend on microbial consortia rather than a single microorganism.

**Keywords:** wash coffee; fermentation; Cesar Department; Tabi variety; Castillo Var.; Colombia Var.







## 1. Introduction

Colombian coffee is known worldwide for its quality. The country produces a mild washed Arabica coffee, which is characterized as a clean cup, with medium/high acidity and body and a pronounced aroma. By 2021, a cultivated area of 844.744 ha was reported, and approximately 92% of Colombian coffee national production was destined for the international market [1]. Coffee is one of the productive sectors that contributes the most to the economic growth of the country. In the 2021/22 coffee year, 11.7 million 60 kg bags of green coffee were produced [2]. With respect to coffee variety growth in Colombia, 70% of the cultivated area corresponds to coffee rust-resistant varieties developed by The National

Coffee Research Center (Cenicafé). In 1982, the variety Colombia (Var. Colombia) was released; this variety is the product of the combination of *Coffea arabica* Var. Caturra, which has a high production level and short stature, and Var. Timor Hybrid, which provides resistance to coffee rust. In 2002, the Tabi variety (Var. Tabi) was developed by crossing the Timor Hybrid with the Típica and Bourbon varieties, both of which are taller trees than Arabica varieties. The Var. Tabi is known for being tall with large beans, resulting in over 80% supreme coffee, and it is considered ideal for obtaining specialty coffees. In 2005, the Castillo General® variety and its regional components were released; these varieties were developed from the cross between the Caturra variety and the Timor Hybrid, resulting in short trees that are adapted to different coffee-growing areas of Colombia and have high-production levels, high resistance to rust and excellent cup quality [3].

In recent years, there has been not only an increase in coffee consumption, but also a new interest in differentiated coffee or coffee with special sensory qualities [4]. Therefore, it is necessary to improve the understanding of postharvest processes that affect coffee quality and thus guarantee a consistent production of high-quality coffee. In addition, there is a need to comply with the market requirements for certified agricultural products with seals that show social and/or environmental commitment. In Colombia, traditionally, the wet processing of coffee is carried out, consisting of pulping, fermentation, washing and drying. During pulping, a machine removes the husk from the coffee fruit, and then the pulped coffee is deposited for a period in a tank to carry out fermentation, of which the purpose is to remove the mucilage attached to the coffee bean or seed. Mucilage is a viscous substance composed of 85% to 91% water and between 6.2% and 7.4% sugars [5]. At this stage, microorganisms, such as bacteria and yeasts, are responsible for degrading these compounds through their metabolism, producing organic acids, alcohols, esters, and volatiles, among other compounds that play an important role as precursors of aromas and flavors in coffee drinks [6–9].

Fermentation processes have been characterized in different coffee-producing countries and in different coffee varieties, such as Var. Típica [10–12], Var. Catuaí [13,14], Var. Caturra [15], Var. Catimor [9], Var. Bourbon [16], Var. Mundo Novo, Ouro Amarelo and Catuaí Vermelho [17] and Geisha [18]. In the vast majority of cases, the predominant microbial groups have been lactic acid bacteria (LAB), such as *Lactobacillus* and *Leuconostoc*, acetic acid bacteria (AAB), *Glucobacter* and *Acetobacter*, and yeasts, such as *Pichia, Candida, Saccharomyces* and *Hanseniaspora*. However, molecular methodologies now allow the identification of many microorganisms that are not cultivable and have revealed great complexity between microbial communities and their dynamics during the fermentation process.

The composition of the microbial community in each fermentation process varies depending on the geographical location, processing method, coffee variety and state of maturity of the coffee fruits at the time of harvest. Factors such as soil, water, tools, insects and human manipulation also influence the types of microorganisms that are abundant during fermentation [9,17,19–21]. These factors also influence the fermentation stage because changes within the same process, such as the consumption of some nutrients and changes in pH and temperature, modify the structures of the microbial population [22]. In addition, the microorganisms metabolic functions can vary according to the interaction with other microorganisms present in the fermentation environment [23]. Although some research has concluded that the presence of microbial species during the fermentation process varies mainly by the region where the process takes place [7,11], the role that coffee variety could play in influencing the microbiological characteristics of fermentation has not been sufficiently addressed.

In Colombia, however, few investigations related to the microbiology of fermentation have been carried out. In Nariño, a study based on the amplification of 16S rDNA demonstrated that LAB, belonging to the genus *Leuconostoc* and the yeast *Pichia nakasei*, predominate throughout the fermentation process. Although less abundant, 54 other microbial genera were identified for the first time in coffee fermentation, mainly associated with edaphic sources, water, and air; these microbes are capable of producing aromatic com-

pounds, enzymes, and organic acids that may be associated with the globally recognized sensory characteristics of Colombian coffee [21]. Likewise, another study in Sierra Nevada de Santa Marta [24] showed that during the fermentation process, bacteria belonging to the genera *Leuconostoc, Acetobacter* and *Latilactobacillus* predominated; within the yeasts, the genus *Kazachstania* predominated, being identified for the first time in coffee mucilage during the fermentation process. However, there is no specific information on the coffee varieties grown in Colombia and their relationship with the composition and microbial dynamics during fermentation.

This work was conducted in the northern region of Colombia on the Sierra Nevada de Santa Marta, where the three aforementioned varieties are currently cultivated. The coffee from this region received the Denomination of Origin from the Colombian Superintendence of Industry and Commerce due to its sensory quality and characteristics associated with the cultural environment and environmental benefits of the region. This coffee is considered a specialty coffee due to the balance of moderate acidity, moderate-large body and sweet chocolate flavor that has resulted in a high-quality and internationally recognized coffee beverage [25]. However, a better understanding of the coffee fermentation processes in this zone is needed, and it is necessary to determine the impact of the microbiome associated with each cultivated coffee variety, the succession of the species during the fermentation process, the biochemical changes caused by microorganisms and how they impact the quality of coffee in the cup.

The objective of this research was to characterize the microbial composition, and physicochemical properties at various fermentation stages of the coffee fermentation process in three varieties of Arabica coffee, namely Var. Colombia, Var. Tabi and Var. Castillo General® and to correlate these microbial compositions and physicochemical properties with the coffee sensory attributes and quality characteristics in cup from the coffee grown in the department of Cesar, representative of Colombian coffee farming.

## 2. Materials and Methods

### 2.1. Location, Sampling, Characterization of the Harvest and Fermentation

The coffee samples were harvested at the Cenicafé, Pueblo Bello Experimental Station (EEPB) in the municipality of Pueblo Bello, Cesar (10°25′19″ north latitude, 73°34′30″ west longitude, 1134 m.a.s.l.), Colombia.

Three coffee varieties, representative of Colombian coffee farming, Var. Colombia, Var. Castillo General and Var. Tabi, were evaluated in December 2021, and the plantations were 6, 8 and 3 years old. To obtain samples, 100 kg of coffee fruits of each variety were manually collected. A sample of water used during wet processing was collected at the beginning of the process in a 500 mL bottle (Schott Duran®), kept in a portable refrigerator (Klimber 27L) and stored at 4 °C until processing for microbiological characterization.

All samples were processed according to the seven practices established by the National Federation of Coffee Growers to reduce the risk of physical and sensory defects in the beverage: 1. ensuring the quality of the coffee harvest, avoiding green and black coffee fruits using the Mediverdes and Cromacafé; 2. processing each batch of coffee independently; 3. classifying and removing inferior quality fruits; 4. cleaning and calibrating the equipment for the benefit of the coffee; 5. monitoring fermentation with Fermaestro®; 6. removing the mucilage with enough water; and 7. obtaining and maintaining coffee parchment with a humidity between 10% and 12% [26].

First, the characterization of the collected coffee fruits was carried out with a Mediverdes® tool (Agroinsumos del café S. A, Bogotá, Colombia) to obtain a representative harvest sample [27], while Cromacafé® (Agroinsumos del café S. A) was used to establish the degree of maturity of the coffee fruit samples [28].

The Mediverdes® and the Cromacafé® are tools that allow us to evaluate the quality of the harvesting by taking a sample of coffee fruits in a 600 mL plastic container and carrying out a classification of stages of fruit maturity with the color chart with 8 stages of maturity, and the percentage of green fruits and the quality of the harvest are thus determined [27,28].

After the coffee fruits were harvested, they were classified by density using the double-basin technique, the floating fruits (i.e. the green and dry fruits, fruits infested with coffee berry borer and empty fruits) were removed, and then coffee pulping was carried out without water in a Gaviota 300 pulper. Then, 1.2 kg of freshly pulped coffee was collected and passed for one min through a BECOLSUB-type experimental mucilage remover (Becolsub DIN-300-Z-NG, JM Estrada S. A, Medellín, Colombia) coupled to Motor Siemens-100422411 (Siemens, Munich, Germany) at 517 rpm with the addition of 900 mL of sterile distilled water, which facilitated the detachment of the mucilage from the seed. At that time, the first sample of mucilage (MT1) was collected, and the beans that passed through the BECOLSUB-type equipment were also collected and served as the first sample of wet coffee seed (BT1), corresponding to zero h of fermentation (T1). The remaining pulped coffee was deposited in an uncovered 500 L plastic tub tank without water added to start the spontaneous fermentation process (wet processing). Additionally, to control the optimal fermentation time, the Fermaestro® (Agroinsumos del café S. A, Bogotá, Colombia) tool was placed in the coffee mass contained in the tank [29]. After 9 and 18 h of fermentation, 1.2 kg of the coffee mass was collected and passed through the mucilage removal equipment as described for T1. The mucilage and seed samples obtained after 9 and 18 h corresponded to time 2 (T2) and time 3 (T3) of sample fermentation, respectively.

The Fermaestro® is a conical plastic tool that allows us to visually identify the washing point when the mucilage removal exceeds 95% of the total mass of pulped coffee, that is, the decrease in the mass volume of coffee must be monitored inside the container to a specific mark that indicates the optimal washing time [29].

For each of the three varieties, the seed samples (ST1, ST2 and ST3) were dried in a parabolic dryer until reaching a humidity between 10 and 12% [30]. The samples were then stored in Ziploc bags and taken for quality analysis to the Quality Laboratory of the Cesar-Guajira Coffee Growers Committee located in the city of Valledupar-Cesar (Colombia). Samples of mucilage (MT1, MT2 and MT3) collected in Nalgene flasks (Thermo Scientific, Waltham, MA, USA) and sterile Falcon tubes were stored at 4 °C in a portable refrigerator (Klimber 27 L) and transferred for preservation at −40 °C and 4 °C at Santander University in Valledupar (Cesar). All mucilage samples were collected in triplicate.

### 2.2. Physicochemical Analysis

At the established times (i.e. T1, T2 and T3) for the collection of mucilage samples, the following parameters were measured. The temperature was measured by introducing a dome thermometer (Brixco, Germany) into the fermentation tank at three distinct points. pH values were determined using indicator paper (Merck MQuant™, Rahway, NJ, USA), employing the same method as for temperature measurement. Lastly, Brix degrees were determined though an optical refractometer (WURTH 0704 510 001) in three replicates of each sample. Data are expressed as the means and SE (standard error) for each of the variables.

#### 2.2.1. Determination of Acidity

The acidity of the MT1, MT2 and MT3 mucilage samples for each variety was determined in triplicate with 50 mL of each sample with the potentiometry technique using 0.1 N (NaOH) solution. The total acidity value was expressed in mg/L equivalent to calcium carbonate ($CaCO_3$) [31].

#### 2.2.2. Determination of Sugars (Sucrose, Glucose and Fructose)

- *Extract preparations*

Extracts of dry green coffee beans were prepared with 0.2 g of cryogenically ground seeds and 10 mL of Milli-Q water, extracting the sugars in a water bath (80 °C) for 30 min, followed by cooling and centrifugation (8.500 r.p.m) (Thermo Scientific™ Megafuge 16R,

USA) for 5 min. One mL of supernatant was taken, filtered through a 0.22 μm membrane and kept at 4 °C until analysis.

Mucilage samples were centrifuged (8.500 r.p.m) for 5 min, and 1 mL of supernatant was filtered through a 0.22 μm membrane and kept at 4 °C until analysis.

- *Sugar analysis*

The samples were injected directly into high-performance liquid chromatography (HPLC) equipment (Waters Alliance 2690, Milford, MA, USA) with a Sugar-Pak I™ (Waters-Millipore, Milford, MA, USA), 6.5 × 300 mm @ 85 °C column (Waters brand). Mobile Phase: Water containing 50 mg/L Ca-EDTA. Flow: 0.5 mL/min- Injection volume: 20 μL. Detector: IR (410) @ 35 °C. Elution mode: Isocratic.

The calibration curve was constructed using the carbohydrate standards relevant to our study. The least square method was employed to analyze the obtained results, leading to determination of both the slope and intercept values. The equation of the line derived from this analysis was subsequently used to calculate the concentration of sucrose, glucose and fructose. These concentrations were expressed as %/g for seed samples and as %/mL for mucilage samples. The results were obtained from data corresponding to 3 lectures from the sample (3 replicates). Means and standard deviations were calculated.

### 2.2.3. Total Protein Quantification

For total protein quantification, 0.1 g of each sample of seed and mucilage previously dried for 18 h in an oven at 90 °C was weighed and packed in tin capsules to be taken directly to the TruSpec® Leco elemental analyzers (St. Joseph, MI, USA) Through complete combustion with pure oxygen, as described by the Dumas method [32], the percentage concentrations of carbon, nitrogen and hydrogen were determined directly with specific detectors for each element. The amount of protein was calculated using the factor of 6.25 [33].

The results were obtained from data corresponding to 3 lectures from the sample (3 replicates). Means and standard deviations were calculated.

### 2.3. Microbiological Analysis

### 2.3.1. Cultivation and Isolation of Microorganisms in Mucilage and Water

The MT1, MT2, and MT3 mucilage samples of each variety were processed for bacterial, yeast, and mycelial fungal determination under the criteria of current colombian regulations [34,35]. Serial dilutions of the coffee mucilage samples were made from 10:1 to 10:4. From each dilution, 100 μL of the coffee mucilage sample was inoculated on Nutrient agar and alate count agar (OXOID, Basingstoke, UK) for mesophilic aerobics and Mac-Conkey agar for enterobacteria, both with Digralsky loops. The Petri dishes were incubated between 26 and 28 °C for 48 h.

For the isolation of lactic acid bacteria (LAB), 5 mL of mucilage was mixed with 25 mL of MRS broth media (MERCK, Darmstadt, Germany) in an Erlenmeyer flask and kept for 24 h at 80 rpm and 29 °C in a Shaker 1000 (Heidolph, Schwabach, Germany). Afterward, the sample was serially diluted to 10:4, and 100 μL of the inoculum of each dilution was spread on MRS media with Digralsky loops. The Petri dishes were incubated between 26 and 28 °C for 48 h.

For the isolation of acetic acid bacteria (AAB), enrichment broths II and III were used (glucose, yeast extract, peptone, ethanol, and broth II containing acetic acid at pH 3.5), and the same contents of broth and mucilage were added to the test tubes. The test tubes were incubated at 30 °C for 72 h at 80 rpm in a Shaker 1000 (Heidolph), and then a 100 μL aliquot was plated on GEY-$CaCO_3$ selective medium (2% glucose, 20% ethanol, 1% yeast extract, 0.3% calcium carbonate ($CaCO_3$) and spread with Digralsky loops on the surface, followed by incubation at 30 °C for 10 days.

Potato dextrose agar (PDA) (OXOID, Basingstoke, UK), yeast extract peptone dextrose agar (YEPD) (MERCK, Darmstadt, Germany), and Sabouraud 4% dextrose agar (SDA) (MERCK, Darmstadt, Germany) were used for the growth of yeast and the mycelial fungus.

Plating was carried out by the surface technique, and plates were incubated at a temperature between 26 and 28 °C for 96 h.

For the microbiological characterization of the water, serial dilutions were carried out to analyze the total and fecal coliforms in Bright Green Broth using the most likely number (MPN) technique with Durham tubes (MERCK, Darmstadt, Germany) according to the regulation [36]. The biomasses of mesophilic aerobes and fungi were also determined, and later, the morphotypes were identified based on the grouping and shape of the macroscopic colonies that developed on the surfaces of the media and the responses of the microscopic colonies to Gram staining [37].

### 2.3.2. Microbial Identification

Mesophiles and enterobacteria: Based on the morphotypes that developed in the established culture media, the groups were identified by macro- and microscopy taxonomic criteria (OLYMPUS CX22LED, Tokyo, Japan). Catalase and oxidase tests and conventional biochemical characterization that included the citrate test I, lysine decarboxylase (LIA), triple sugar agar (TSI), phenylalanine (FA), urea, mobility, indole, $H_2S$ and gas (SIM). Methyl red and the Voges–Proskauer (RM/VP) test and nitrate broth were used to identify the groups of fermenting bacteria belonging to the Enterobacteriaceae family. Additionally, the identifications were confirmed with a 96-E IST card panel (BIOTECH, CO., Carrollton, TX, USA).

Lactic acid bacteria: Each isolate colony was cultured in enrichment MRS broth at 29 °C for 24 h at 80 rpm. After incubation, serial dilutions were made in MRS broth up to 10:3, and 100 µL was seeded on MRS agar by spreading on the surface with a Drigalski loop. The plates were incubated at 30 °C for 24 to 48 h. The count and characterization of each morphotype was carried out by macroscopic observation techniques and microscopy (OLYMPUS CX22LED, Tokyo, Japan). In addition, biochemical tests were carried out by the API 50CH and API 50CHL methods (Biomerieux®, Biotechnology company, Minato City, Japan).

Acetic acid bacteria: Enrichment and isolation were carried out in two culture media: MII (glucose, peptone, ethanol, yeast extract, and acetic acid) and MIII (glucose, ethanol, yeast extract, peptone) at pH 3.5 and 4.5, respectively [38]. The solutions were incubated at 30 °C for 72 h under aerobic conditions, and then a 100 µL aliquot was inoculated in solid GEY-CaCO$_3$ medium with the addition of 0.3% spiramycin 3 M.U. I (Labinco, Breda, The Netherlands) and incubated at 30 °C for 10 days. Then, the characteristics of the morphotype and formation of a translucent halo around the cfu were observed, and the colony was stained with Gram stain; in addition, catalase and cytochrome oxidase tests were performed [39].

Yeasts: The isolated morphotypes were subjected to phenotypic characterization by means of lactophenol blue staining and observation of their microscopic characteristics. Later, the isolates were identified using the DL-96II ID/AST microbial system to identify the yeasts by colorimetry and susceptibility by turbidimetry semiquantitative analysis of antimicrobial minimum inhibitory concentrations (MICs) following the manufacturer's instructions (Zhuhai DL Biotech Co., Ltd., Zhuhai, China).

Mycelial Fungi: Morphotypes were identified from the development of colonies on PDA (OXOID, Basingstoke, Hampshire, UK), Sabouraud 4% dextrose agar (Merck) and YEPD agar (Sigma Aldrich, Burlington, MA, USA). Macroscopic identification of the colonies (front and back) as well as their appearance (shape, size, color and texture) was performed by reviewing taxonomic keys [40]. Microscopic characterization was carried out through the microculture technique to observe the reproductive structures typical of a fungus and to facilitate identification with a taxonomic key of imperfect fungi [41].

### 2.4. Molecular and Metataxonomic Analyses

Molecular and metataxonomic analyses were carried out at the National Center for Genomic Sequencing (NCGS), Universidad de Antioquia, Medellin, Colombia. DNA

extraction was performed on the MT1, MT2 and MT3 mucilage samples for each variety. For DNA extraction, 2 mL of coffee mucilage was collected, and centrifugation was carried out at $16,000 \times g$ for 5 min. The sediment was used to extract genomic DNA (gDNA) with the QIAGEn DNeasy Powerlyzer Powersoil Kit (Hilden, Germany). At the end of the extraction process, DNA quantification was performed by the light absorption method at 260 nm using a NanoDrop™ 2000 (Thermo Scientific™, Waltham, MA, USA). The obtained gDNA samples were frozen at $-20$ °C for the analysis of the microbial diversity of bacteria, yeasts and fungi.

gDNA extracted from the samples was normalized to a concentration of 30 ng/μL. Subsequently, Illumina 300 bp paired libraries were prepared and sequenced at Macrogen Inc. following the recommendations (Seoul, Republic of Korea).

The DNA samples were used to determine bacterial metataxonomic diversity by PCR amplification of the 16S rDNA molecular marker and the V3 and V4 variable regions using the primers Bakt_341F (CCTACGGGNGGCWGCAG) and Bakt_805R (GAC-TACHVGGGTATCTAATCC). Additionally, fungal diversity was determined using the ITS molecular marker and ITS3F (GCATCGATGAAGAACGCAGC) and ITS4R (TCCTCCGCT-TATTGATATGC) primers [42].

Deep sequencing was performed on the Illumina MiSeq platform, generating paired reads of 300 bases each. The reads were cleaned to a Q30 quality threshold, and singletons and sequences shorter than 200 bases in length were eliminated using the program Cutadapt version 3.5. Sequences were analyzed using the Mothur program version 1.44 following the standard protocol for Illumina MiSeq libraries (SOP) [43]. Paired-end (PE) reads were assembled using the Mothur make.contigs tool and then aligned to the 16S Silva reference database (Silva.nr v138). Subsequently, the VSEARCH algorithm was used to remove the chimeric sequences. Sequences from nonbacterial lineages (sequences of mitochondrial, chloroplast, archaeal, and eukaryotic origin) were removed for further analysis. The dist.seqs routine was used to group the reads into operational taxonomic units (OTUs), considering a distance limit between sequences of 0.03. The data were normalized with the normalize.shared command. The phylogenetic classification of the OTUs for fungi and bacteria with taxonomic assignment at the family and genus levels for both bacteria and fungi was carried out with the 16S Silva database (Silva.nr v138) with a threshold of 80 (80 bootstrap threshold) using the RDP Classifier algorithm. Coverage analysis was performed to determine the sequencing coverage using Mothur, and the rarefaction curve was also analyzed.

With the sequencing results of each sample, the microbial diversity: alpha diversity indices were calculated, including the ACE, Chao1, Observed, Simpson and Shannon indices, and beta diversity (nonmetric multidimensional scaling-NMDS) was calculated with the Phyloseq and Microbiome packages of the RStudio program (version 2022.07.0). Statistical tests and graphs were carried out with the same program.

### 2.5. Physical and Sensory Analyses of Coffee

The analysis of the physical quality of the green coffee beans was based on the determination of humidity, the percentage loss, low grade, black and vinegar beans, coffee berry borer-infested grains and the percentage of healthy beans according to Instituto Colombiano de Normas Técnicas y Certificación (2021) [44].

The sensory evaluation was carried out according to the protocol described by the specialty coffee association (SCA) [45] and with the participation of 3 cuppers who are certified as Q Graders by the Coffee Quality Institute (CQI) and belong to the sensory panel of the Quality Laboratory of the Cesar-Guajira Coffee Growers Committee. Based on this protocol, ten coffee attributes were registered: fragrance/aroma, flavor, residual flavor, acidity, body, balance, uniformity, clean cup, sweetness, taster score, defects, and total. The sensory quality, expressed as the SCA total score, was the response variable.

## 3. Results

### 3.1. Characterization of the Harvest

The Mediverdes® tool was used for the harvesting evaluation of the three varieties tested, and the harvests were categorized as excellent, with percentages of green fruits less than 0.38% (Data available in the Supplementary Material Table S1).

### 3.2. Physicochemical Analysis

3.2.1. Temperature, pH, °Brix and Acidity in the Mucilage

The temperature during the fermentation process showed an increase for all varieties, with an initial average of 24 °C and increases between 1 and 4 °C at the end of the process. The pH decreased, with an initial average value of 4.29 and a value of 3.62 at the end of fermentation. The total acid content tripled relative to the initial content in all evaluated processes; the acid content of Var. tabi increased from 315.07 to 1015.07 mg/L $CaCO_3$; that of Var. Castillo General increased from 188.40 to 748.40 mg/L $CaCO_3$, and that of Var. Colombia increased from 198.40 to 655.07 mg/L $CaCO_3$. On the other hand, the changes in the Brix degrees varied with variety: the highest values were obtained for Var. Colombia, increasing from 5.17 to 6.83° Brix during fermentation, before decreasing to 6.00° Brix at the end of the fermentation process. For Var. Castillo General and Var. Tabi, values of 5.00 and 4.00° Brix were observed at the beginning of the process, respectively. At the end of the fermentation period, the Brix value of Var. Tabi increased to 4.83° Brix, while that of Var. Castillo General decreased to 4.43° Brix (Table 1).

**Table 1.** Physical–chemical analysis of coffee mucilage during the fermentation process at the Pueblo Bello Experimental Station. ± standard error.

| Sample | Time (h) | Temperature (°C) | pH | °Brix (°Bx) | Total Acidity (mg/L $CaCO_3$) |
|---|---|---|---|---|---|
| Var. Tabi | 0 | 25 ± 1.50 | 4.05 ± 0.16 | 4.00 ± 0.01 | 315.07 ± 15.27 |
| | 9 | 28 ± 1.04 | 3.83 ± 0.15 | 2.00 ± 0.01 | 405.07 ± 11.54 |
| | 18 | 29 ± 0.50 | 3.83 ± 0.16 | 4.83 ± 0.28 | 1015.07 ± 28.86 |
| Var. Castillo General | 0 | 24 ± 1.50 | 4.38 ± 0.54 | 5.00 ± 0.01 | 188.40 ± 17.32 |
| | 9 | 27 ± 0.76 | 4.00 ± 0.22 | 5.00 ± 0.50 | 448.40 ± 17.32 |
| | 18 | 26 ± 1.00 | 3.56 ± 0.06 | 4.43 ± 0.40 | 748.40 ± 50.00 |
| Var. Colombia | 0 | 24 ± 0.50 | 4.44 ± 0.65 | 5.17 ± 0.28 | 198.40 ± 86.60 |
| | 9 | 25 ± 0.50 | 4.13 ± 0.18 | 6.83 ± 0.28 | 288.40 ± 17.32 |
| | 18 | 26 ± 0.50 | 3.49 ± 0.07 | 6.00 ± 0.01 | 655.07 ± 211.26 |

± standard error.

The organic acid contents in the mucilage (Table 1) increased during the fermentation process for the three varieties evaluated. The highest total acidity at the beginning of the process was observed in var. Tabi. This one was also the one that showed the highest total acidity increase from T1 to T3 (the increment was 700 mg/L), followed by Var. Castillo, which showed an increase of 560 mg/L from the beginning to the end of the fermentation process. Var Colombia was the one with the lowest differences in total acidity, with a 124 mg/L increment from the beginning to the end of the fermentation process.

3.2.2. Sugar and Protein Contents

The sucrose contents in the mucilage during the fermentation process were between 0.01 and 0.44% and were much lower than the contents in the seeds. In Var. Tabi, the changes were few. In Var. Colombia, at the beginning of the process, the sucrose content was 0.07%, but at the middle and final times, this sugar was not detected; on the other hand, Var. Castillo General showed an increase from 0.17 to 0.44% from the beginning to the middle of the fermentation period; however, at the end of the process, this value decreased to 0.05% (Table 2). In the seeds, the sucrose contents were between 6.70 and 8.54%.

**Table 2.** Sugar and protein contents in mucilage and coffee beans of three varieties during the fermentation process.

| Sample | Time (h) | Sucrose | | Glucose | | Fructose | | Protein | |
|---|---|---|---|---|---|---|---|---|---|
| | | **M** | **S** | **M** | **S** | **M** | **S** | **M** | **S** |
| | | % | | | | | | | |
| Var. Tabi | 0 | 0.04 ± 0.01 | 8.13 ± 0.30 | 0.85 ± 0.32 | ND | 1.43 ± 0.38 | 0.23 ± 0.01 | 10.31 ± 1.84 | 13.94 ± 0.01 |
| | 9 | ND | 7.52 ± 0.22 | 0.20 ± 0.11 | 1.32 ± 0.38 | 0.67 ± 0.24 | 0.22 ± 0.01 | 14.00 ± 1.15 | 13.94 ± 0.05 |
| | 18 | 0.02 ± 0.01 | 7.24 ± 0.13 | 0.07 ± 0.05 | 1.39 ± 0.03 | 0.51 ± 0.07 | 0.22 ± 0.01 | 14.56 ± 0.28 | 14.13 ± 0.09 |
| Var. Castillo General | 0 | 0.17 ± 0.13 | 7.29 ± 0.39 | 0.40 ± 0.04 | 1.42 ± 0.04 | 1.58 ± 0.15 | 0.17 ± 0.09 | 9.19 ± 1.21 | 14.69 ± 0.09 |
| | 9 | 0.44 ± 0.19 | 8.08 ± 0.22 | 1.37 ± 0.02 | 1.51 ± 0.03 | 1.88 ± 0.08 | 0.36 ± 0.05 | 11.63 ± 0.61 | 14.50 ± 0.04 |
| | 18 | 0.05 ± 0.01 | 7.30 ± 0.38 | 0.83 ± 0.26 | 1.39 ± 0.06 | 1.87 ± 0.01 | 0.13 ± 0.11 | 10.63 ± 0.50 | 14.63 ± 0.06 |
| Var. Colombia | 0 | 0.07 ± 0.03 | 6.87 ± 0.08 | 2.43 ± 0.43 | 1.25 ± 0.01 | 2.93 ± 0.48 | 0.23 ± 0.08 | 11.19 ± 0.56 | 14.94 ± 0.25 |
| | 9 | ND | 6.70 ± 0.50 | 1.55 ± 0.30 | 1.25 ± 0.03 | 1.97 ± 0.39 | 0.06 ± 0.04 | 10.06 ± 0.69 | 14.44 ± 0.13 |
| | 18 | ND | 8.54 ± 0.91 | 0.31 ± 0.08 | 1.38 ± 0.06 | 1.44 ± 0.26 | 0.15 ± 0.04 | 12.81 ± 1.37 | 14.56 ± 0.06 |

M: Mucilage. S: Seed. ND: Not detected. ± standard error.

The percentage of glucose in the mucilage at the beginning of fermentation (T1) was between 0.40 and 2.92%, and in Var. Tabi and Var. Colombia, there was a decrease in the glucose content at the end of the fermentation period, with percentages of 0.05 and 0.31%, respectively. The glucose content of Var. Castillo General increased slightly, from 0.40% to 0.83%, at the end of the fermentation period.

In the seeds, the glucose percentages were similar to those observed in the mucilage. At the beginning of the fermentation period, glucose was not observed in Var. Tabi; however, midway through the fermentation process, the level increased to 1.32%, and at the end of fermentation, a content of 1.39% was recorded. This value was very similar to that obtained for Var. Castillo General and Var. Colombia, with percentages of 1.39 and 1.38%, respectively. No variations in the seed glucose content were observed during the fermentation period for Var. Castillo General and Var. Colombia.

The fructose contents in the mucilage of Var. Tabi and Var. Colombia decreased from the beginning to the end of fermentation, with values of 1.43 to 0.51% and 2.93% to 1.44%, respectively. A slight increase in the fructose content was observed in Var. Castillo General from the start (1.58%) to the end (1.87%) of fermentation. In the seeds, the variations in the fructose content were lower than those observed in the mucilage, with values between 0.06 and 0.36%.

The protein content in the mucilage ranged between 9.1 and 14.5%, and during the fermentation process, a slight increase was observed relative to that at the initial evaluation time for the three evaluated varieties. In contrast, in the seeds, no differences were observed between varieties or between evaluation times, with an average protein content of 14.4% (Table 2).

### 3.3. Microbiological Analysis of Coffee Mucilage and Water

The microbial biomass in the mucilage samples of the three varieties is shown in Table 3. Microbial counts were conducted for the different groups evaluated. For the mesophiles in Var. Tabi, Var. Castillo General, Var. Colombia and at the start of the fermentation process, the average logarithmic concentrations were 5.89, 5.90 and 5.92 log10 cfu/mL, and after 18 h of fermentation, there was a slight decrease to 5.47, 5.48 and 5.57 log10 cfu/mL, respectively. For the three varieties, the counts were 3.60, 3.48 y 4.00 log10 cfu/mL for coliforms and 5.77, 5.91 and 5.44 log10 cfu/mL for yeasts, and there was a reduction in the load for all three varieties. The development of mycelial fungi increased in Var. Tabi, while in the Castillo General and Colombia varieties, growth decreased or was maintained at a concentration of 3.30 log10 cfu/mL. The LAB showed the highest count at the beginning of the fermentation process, with a logarithmic average of 6.77 log10 cfu/mL, indicating a decrease in the population by the end of the process for the three varieties of coffee.

**Table 3.** Microbial population present in the mucilage of three varieties of Arabica coffee during 0, 9 and 18 h of fermentation.

| Sample | Time (h) | Microbial Counts | | | | |
| --- | --- | --- | --- | --- | --- | --- |
| | | Mesophiles | Coliforms | Acid Lactic Bacteria | Yeasts | Mycelial Fungi |
| | | Log10 cfu/mL | | | | |
| Var. Tabi | 0 | 5.89 | 3.60 | 6.78 | 5.77 | 3.30 |
| | 9 | 5.61 | 3.48 | 6.45 | 5.58 | 3.00 |
| | 18 | 5.47 | 3.00 | 4.58 | 5.48 | 3.48 |
| Var. Castillo General | 0 | 5.90 | 3.48 | 6.77 | 5.91 | 3.30 |
| | 9 | 5.32 | 3.30 | 4.07 | 5.24 | 3.00 |
| | 18 | 5.48 | 3.00 | 6.30 | 5.38 | 3.30 |
| Var. Colombia | 0 | 5.92 | 4.00 | 6.76 | 5.44 | 3.48 |
| | 9 | 5.66 | 3.60 | 4.04 | 5.36 | 3.00 |
| | 18 | 5.57 | 3.30 | 6.38 | 5.24 | 3.30 |

Table 4 shows the bacteria isolated from the coffee mucilage samples evaluated during fermentation, highlighting the LAB group, especially the genera *Lactobacillus*, *Leuconostoc* and *Lactococcus* and the species *Lactiplantibacillus plantarum*. In the group belonging to the Enterobacteriaceae family, the presence of *Enterobacter*, *Shigella*, *Citrobacter*, *Escherichia*, *Averyella* and *Klebsiella ozaenae* as well as the genera *Micrococcus*, *Staphylococcus*, and *Bacillus* and the species *Bacillus firmus* were noted. Among the AAB, the genus *Gluconobacter* sp. was observed in the mucilage of the Castillo General and Tabi varieties.

**Table 4.** Identification of bacteria present in the coffee mucilage of the three varieties during the fermentation process.

* LAB: Lactic Acid bacteria; ** AAB: Acid Acetic Bacteria.

Regarding the development of yeasts and mycelial fungi (Table 5), the presence of yeasts and mycelial fungi in the three varieties at the three sampling times stood out, and

the genera and species with the highest frequencies were *Saccharomyces cerevisiae* in Var. Tabi and Var. Colombia, *Candida krusei* in the Castillo General and Colombia varieties and *Pichia* in var. Tabi and Castillo General. Less frequently, *Candida parapsilosis*, *C. rugosa*, *Cryptococcus laurentii*, *C. neoformans* and *Rhodotorula* were observed. The development of filamentous fungi, such as *Trichoderma, Geotrichum* and *Penicillium*, was evidenced in the Tabi and Castillo General varieties; in the Colombia variety, the genus *Geotrichum sp.* was not found.

**Table 5.** Identification of yeasts and mycelial fungi present in the mucilage of three varieties of Arabica coffee during the fermentation process.

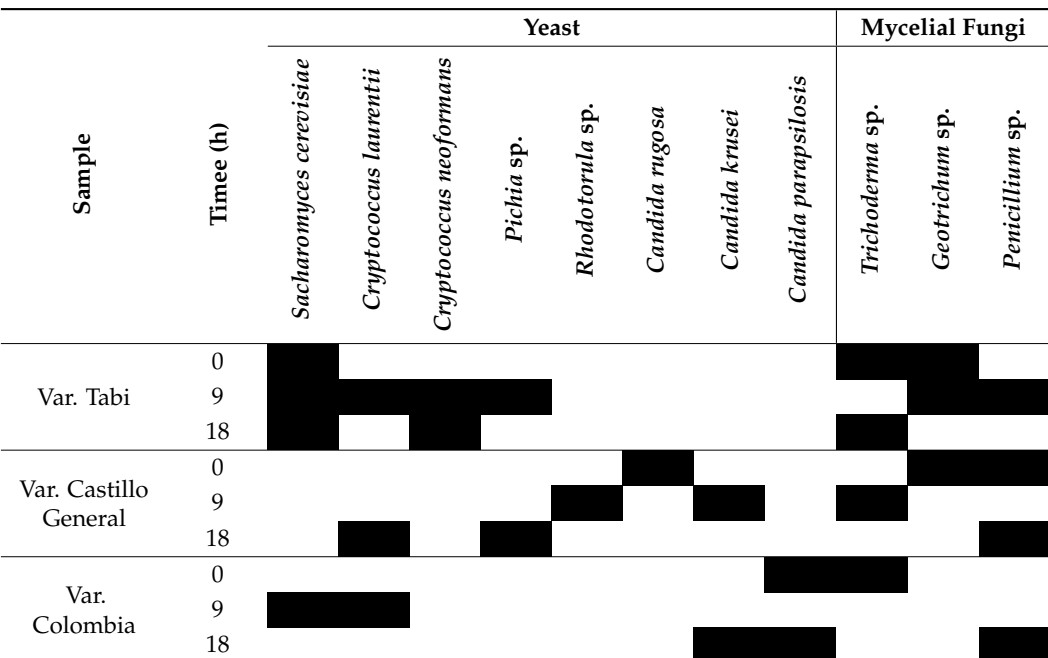

| Sample | Timee (h) | Sacharomyces cerevisiae | Cryptococcus laurentii | Cryptococcus neoformans | Pichia sp. | Rhodotorula sp. | Candida rugosa | Candida krusei | Candida parapsilosis | Trichoderma sp. | Geotrichum sp. | Penicillium sp. |
|---|---|---|---|---|---|---|---|---|---|---|---|---|
| | | | | | | **Yeast** | | | | | **Mycelial Fungi** | |
| Var. Tabi | 0 | ■ | | | | | | | | ■ | ■ | |
| | 9 | ■ | ■ | ■ | ■ | | | | | | ■ | ■ |
| | 18 | ■ | | ■ | | | | | | ■ | | |
| Var. Castillo General | 0 | | | | | | ■ | | | | ■ | ■ |
| | 9 | | | | | ■ | | ■ | | ■ | | |
| | 18 | | ■ | | ■ | | | | | | | ■ |
| Var. Colombia | 0 | | | | | | | | ■ | | | |
| | 9 | ■ | ■ | | | | | | | | | |
| | 18 | | | | | | | ■ | ■ | | ■ | |

The microbiological characterization of the water available for use in coffee wet processing showed that the water contained 29 NMP/100 mL total coliforms and thermo-tolerant coliforms. The taxa identified included *Escherichia coli*, *Proteus vulgaris*, *Enterobacter* sp. and *Pantoea* sp. The count of mesophilic bacteria was 3.19 (log10 cfu/mL) and that of yeasts was 2.69 log10 cfu/mL. The water used in the wet processing of coffee did not achieve the sanitary conditions required for human consumption [37].

*3.4. Molecular and Metataxonomic Analyses*

The rarefaction curves (data available in the Supplementary Material Figures S1 and S2) and coverage analysis for the processed samples showed a coverage value >97% for bacteria and >99% for fungi (data available in the Supplementary Material Tables S2 and S3), indicating that the sampling was sufficient and representative of the diversity in the samples of interest. Based on these samples, OTUs were generated.

The bacterial alpha diversity in the coffee mucilage of the three varieties was determined during the fermentation process (Table 6).

The number of species observed (SObs) in the three varieties was higher at the beginning of the fermentation process than at the end of the fermentation process. The ACE richness index in Var. Castillo General (1516.2) was much higher than that in the Tabi (420.1) and Colombia (505.1) varieties at the start of the process; the CHAO1 indices at the beginning of fermentation varied among the varieties: Var. Tabi had the lowest value of 3.958, which decreased to 2.646 by the end of the process. A decrease was also observed at the end of fermentation in the Colombia and Castillo General varieties, which had similar CHAO1 values of 3.771 and 3.627, respectively.

**Table 6.** Bacterial diversity during the fermentation of three coffee varieties.

| Sample | Time (h) | Sobs * | Richness | | Diversity | |
|---|---|---|---|---|---|---|
| | | | Ace | Chao1 | Simpson | Shannon |
| Var. Tabi | 0 | 266 | 420.170 | 3.958 | 2.677 | 0.879 |
| | 9 | 224 | 366.067 | 3.711 | 2.196 | 0.820 |
| | 18 | 177 | 259.307 | 2.646 | 2.062 | 0.804 |
| Var. Castillo G | 0 | 570 | 1516.287 | 13.276 | 3.292 | 0.894 |
| | 9 | 194 | 318.577 | 3.130 | 2.113 | 0.811 |
| | 18 | 172 | 355.041 | 3.627 | 2.298 | 0.852 |
| Var. Colombia | 0 | 314 | 505.159 | 4.958 | 1.862 | 0.589 |
| | 9 | 238 | 698.933 | 5.579 | 1.931 | 0.754 |
| | 18 | 183 | 361.125 | 3.771 | 2.295 | 0.850 |

* Species Observed.

However, in all cases, the richness values decreased between the start and end of fermentation. The Simpson and Shannon diversity indices were similar in Var. Tabi and Var. Castillo General; in both cases, there was a decrease in the values at the end of fermentation. In contrast, in Var. Colombia, there was an increase in these indices during fermentation; however, the final values were similar among the three varieties.

Figure 1 shows the multidimensional nonmetric scaling (NMDS) analysis results of the three varieties according to bacterial and fungi diversity. The results indicate that the samples of the Castillo General and Colombia varieties had similar bacterial microbiota, while in Var. Tabi, greater differences were observed due to distant clustering of the three samples. The NMDS results of fungi and yeasts during the fermentation process for the three coffee varieties did not show the greatest differences in the fungal and yeast populations, but as observed for bacteria, the Var. Castillo data were the most dispersed.

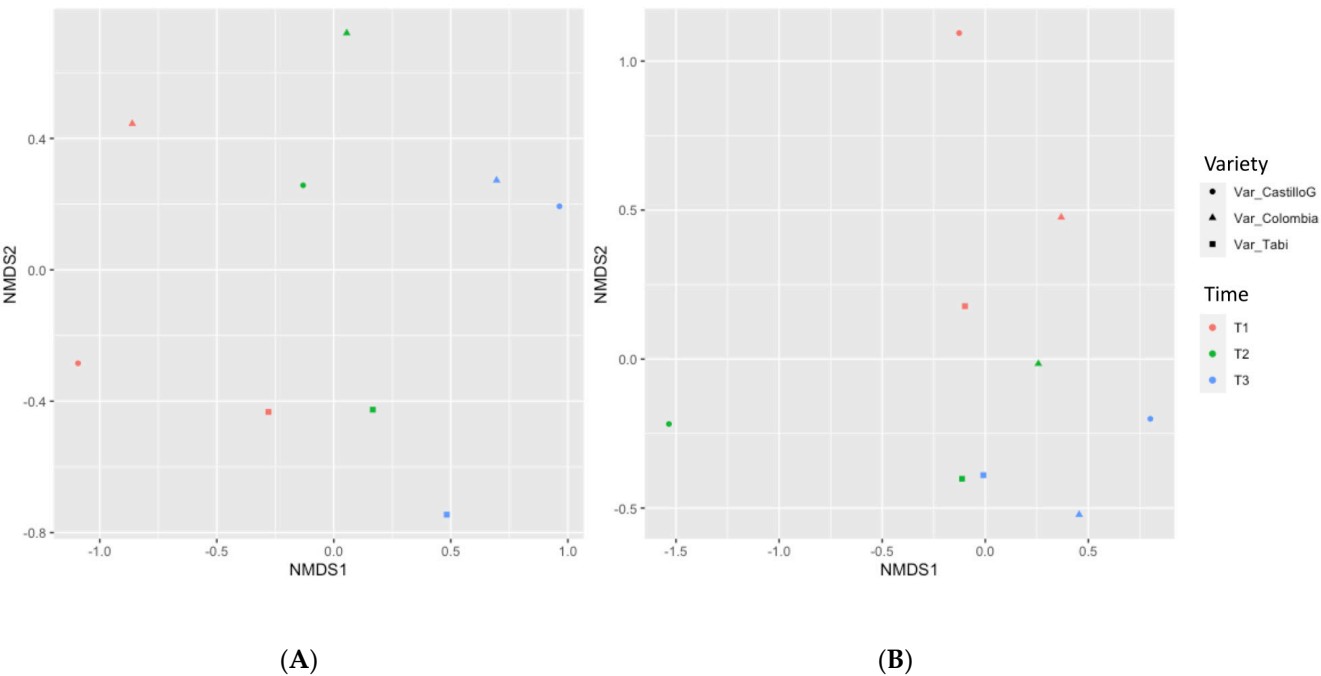

**(A)**                                                        **(B)**

**Figure 1.** NMDS analysis of (**A**) Bacteria and (**B**) Fungi associated with the three varieties evaluated.

Figure 2 shows the top 10 taxonomic assignments at the family level in the bacterial communities associated with the three varieties evaluated during the fermentation process. In Var. Tabi, the family *Acetobacteraceae* was predominant, followed by *Leuconostocaceae*. The population (OTUs) of these two families increased during fermentation, while the Enterobac-

teriaceae family decreased during fermentation. In the case of Var. Castillo General and Var. Colombia, although the microbial communities differed from the beginning to the end of the fermentation process, the same families (*Acetobacteraceae, Enterobacteriaceae* and *Leuconostocaceae* sp.) and similar proportions of those families were observed for both varieties. There is an assignment identified as other; in this, the family *Lactobacillaceae* is present as was observed in the 16S.trim.contigs.good.unique.good.fileter.uniquaprecluster.pick.nr_v132.wang.tax.summary. In addition, genus from this family were isolated into the pure culture in Var. Colombia.

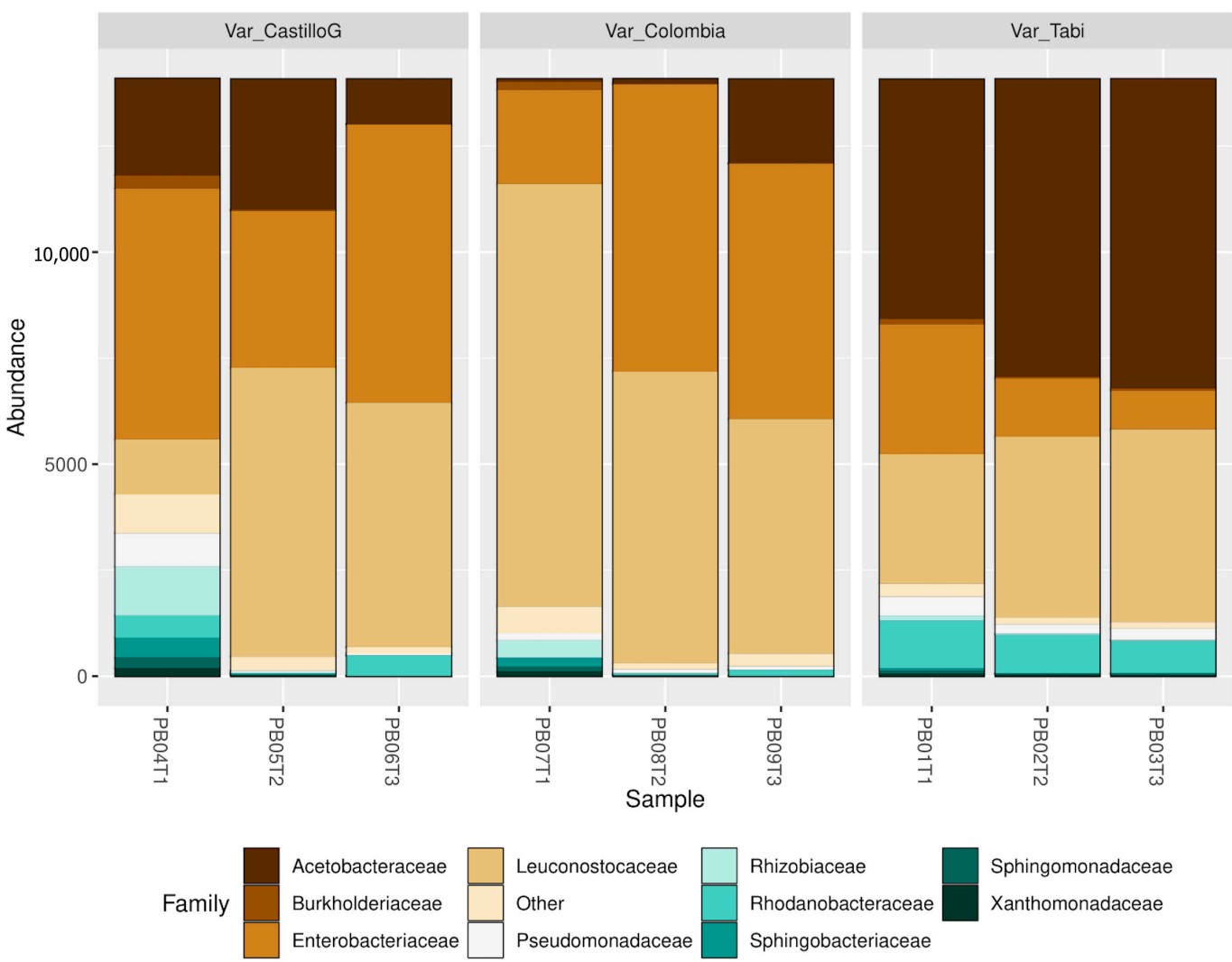

**Figure 2.** Taxonomic assignment of bacteria in the varieties analyzed at the family level.

Figure 3 shows the top 10 taxonomic assignment at the genus level; the results indicated that the main bacterial genera associated with Var. Tabi were *Gluconobacter, Leuconostoc, Acetobacter, Frateuria* and unclassified genera of the Enterobacteriaceae group, with slight changes in the proportions of these genera during the fermentation process. Unclassified bacteria from the Enterobacteriaceae group were the primary taxa associated with Var. Castillo General at the start of the fermentation period, followed by *Gluconobacter* sp., *Leuconostoc* sp., *Pantoea* sp., *Pseudomonas* sp., *Tatumella* sp. and an unclassified *Rhizobiaceae* sp. At the midway point in the fermentation process, a large increase in the abundances of *Leuconostoc* sp., *Gluconobacter* sp. and *Pantoea* sp. was observed; however, the population of these bacteria decreased again at the end of the fermentation period, when large proportions of bacteria belonging to the genera *Tatumella* and *Weissella* were observed.

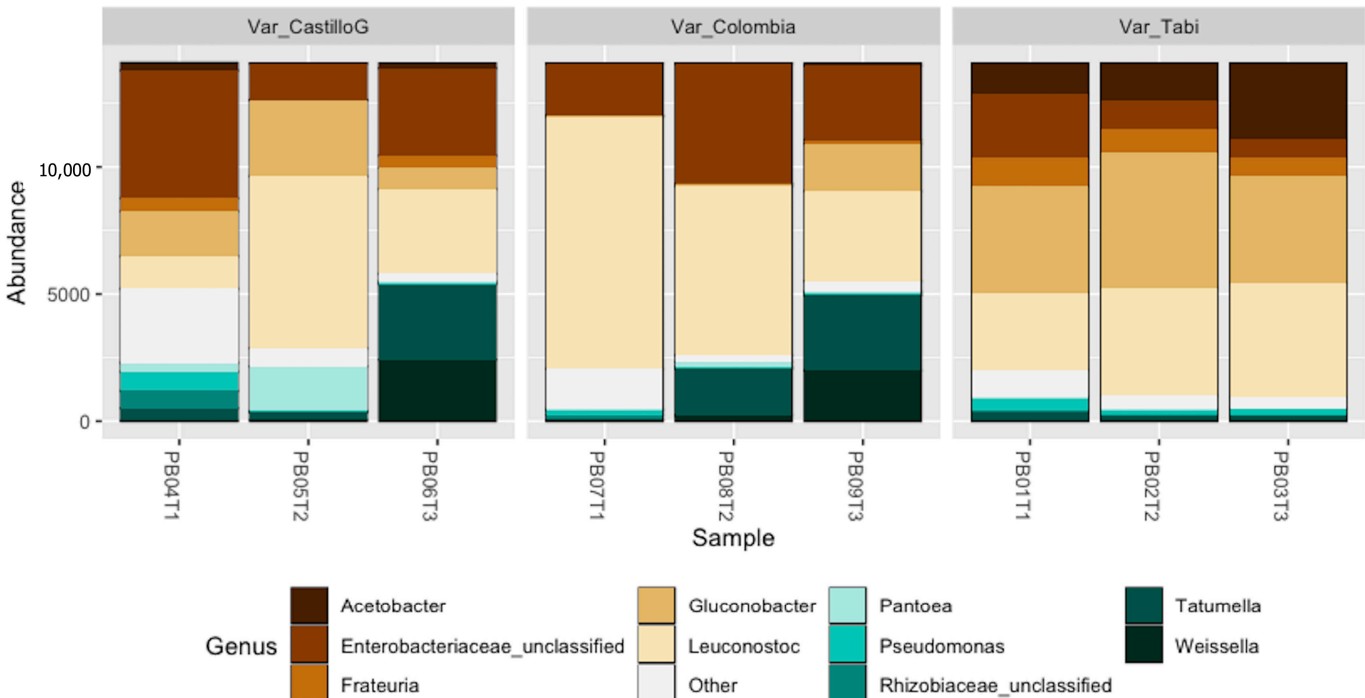

**Figure 3.** Taxonomic assignment of bacteria associated with the varieties analyzed at the genus level.

Again, it is noteworthy that at the end of the fermentation process, the groups of mixed acid bacteria (*Enterobacteriaceae*) and LAB (*Leuconostoc* sp.) were similar between the Castillo General and Colombia varieties, while in Var. Tabi, there was an equal abundance of members of the AAB group (*Acetobacter* sp. and *Gluconobacter* sp.) and LAB group. In addition, an assignment identified as "others" is observed, and in this one, different genus from the Familia *Lactobacillaceae* were present.

The alpha diversity of fungi in the coffee mucilage of the three varieties was assessed during the fermentation process (Table 7). The number of observed species was higher at the beginning of the fermentation process than at the end of the fermentation process, which was similar to what was observed for the bacterial communities. Similarly, the highest values of richness and diversity were observed for Var. Castillo General at the start of the fermentation period; these values were higher than those observed for the Tabi and Colombia varieties. However, in all varieties, the richness and diversity of the fungal communities decreased during fermentation.

**Table 7.** Fungal diversity during fermentation of three coffee varieties.

| Sample | Time (h) | Sobs * | Richness | | Diversity | |
|---|---|---|---|---|---|---|
| | | | Ace | Chao 1 | Simpson | Shannon |
| Var. Tabi | 0 | 574 | 446.570 | 4.653 | 2.167 | 0.782 |
| | 9 | 436 | 286.931 | 2.644 | 1.700 | 0.685 |
| | 18 | 419 | 383.130 | 3.418 | 1.684 | 0.685 |
| Var. Castillo G | 0 | 543 | 1756.051 | 29.155 | 2.519 | 0.849 |
| | 9 | 227 | 577.207 | 4.810 | 2.406 | 0.842 |
| | 18 | 215 | 529.321 | 4.356 | 1.445 | 0.612 |
| Var. Colombia | 0 | 577 | 389.022 | 3.703 | 1.858 | 0.679 |
| | 9 | 524 | 359.125 | 3.017 | 2.028 | 0.768 |
| | 18 | 320 | 212.395 | 2.074 | 1.316 | 0.601 |

* Species Observed.

Figure 4 shows the top 10 taxonomic assignments at the family level of the fungal community present during the fermentation of the three coffee varieties. The *Saccharomycodaceae* family accounted for a large proportion of the identified fungi, except in Var. Castillo General, in which the *Saccharomycetaceae* family predominated; this family was not particularly abundant in the other varieties. *Phaffomycetaceae* was more abundant at the three evaluation times in Var. Colombia than in Var. Castillo General or Var. Tabi. In Var. Tabi, there were no large changes in the fungal community across the evaluation times, and the predominant fungi belonged to the *Saccharomycodaceae* family, followed by *Pichiaceae*. Once again, the Castillo General and Colombia varieties were associated with the same families (*Phaffomycetaceae* and *Phaffomycetaceae*) in similar proportions at the end of the fermentation period.

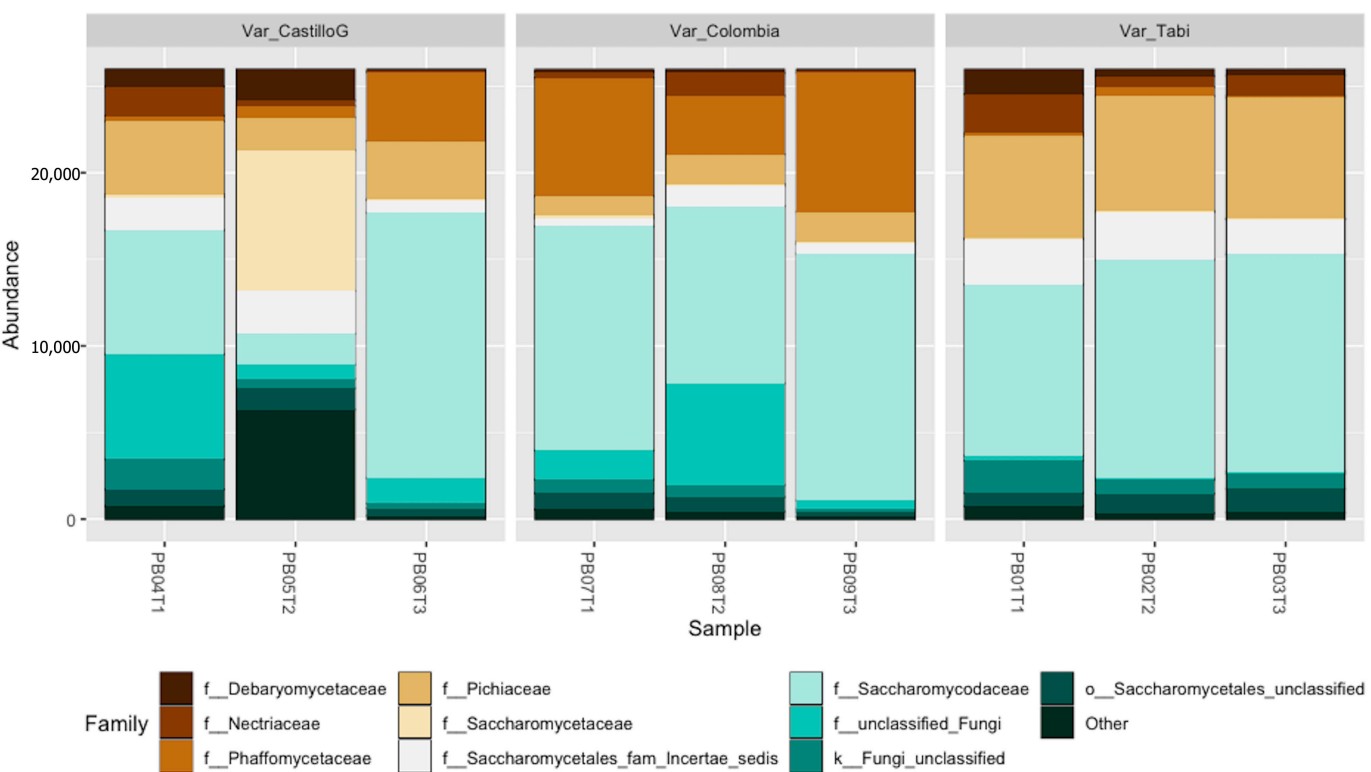

**Figure 4.** Taxonomic assignment of fungi associated with the varieties analyzed at the family level.

Figure 5 shows the top 10 genera in the fungal communities during fermentation. In Var. Tabiar. Colombia and Var. Castillo General, unidentified genera from Saccharomycodaceae were predominant. In addition, in Var. Tabi, the presence of *Candida* and *Pichia* spp. was observed, while in Var. Colombia, *Wickerhamomyces* was widely present. On the other hand, in Var. Castillo Genera, the presence of *Zygotorulaspora* spp. Pichia spp. and *Penicillium* spp. was observed.

### 3.5. Physical and Sensory Analyses

The values of the physical variables for all seed samples were within the ranges considered normal [44]. The seed samples had a humidity between 10.6 and 12.10%. The loss accounted for between 15.3 and 17% of the sample; black and vinegar grains accounted for between 0 and 0.33%; grains infested with coffee berry borer accounted for between 0.02 and 5.2%; and healthy grains accounted for between 77 and 81%. The performance factor of the grains was between 85.6 and 90%. The specific data for each evaluation are not shown.

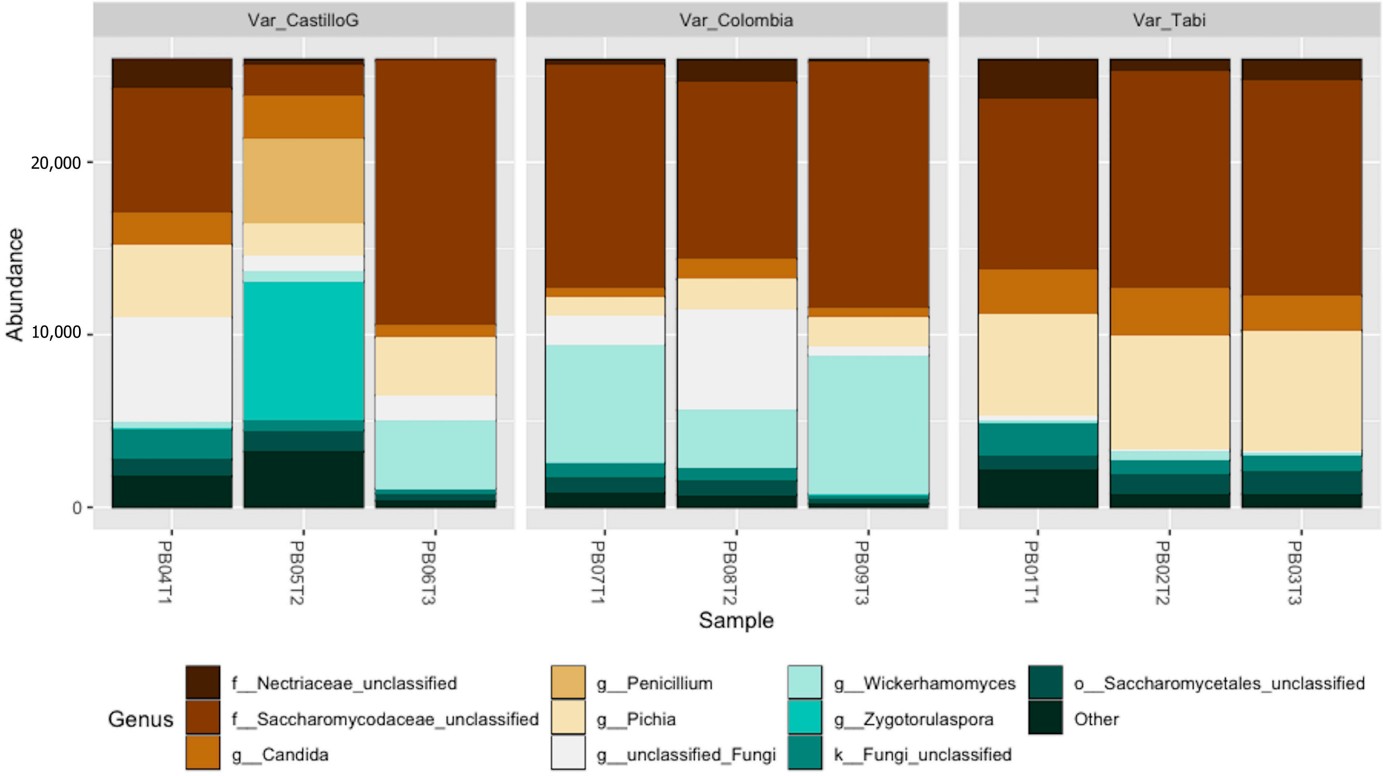

**Figure 5.** Taxonomic assignment of fungi associated with the varieties analyzed at the genus level.

The SCA scores are shown in Table 8, and changes can be seen among the sampling times and the evaluated varieties. The Tabi and Colombia varieties had little variation and similar scores. The Tabi variety had an initial score of 82.8 and a final score of 82.7, while Var. Colombia had an initial score of 82.2 and a final score of 82.5. Var. Castillo General had an initial score of 53.63, which decreased to 52.0 at the end of the fermentation period. In Var. Castillo General, the low score was related to a storage defect at the initial and final evaluations. The storage defect was not related to the fermentation process; it is possible that the presence of this defect was associated with temperature changes during the storage of the samples after the drying process.

**Table 8.** SCA scores of three coffee varieties at different times in the fermentation process.

| Sample | Time (h) | SCA | Sensory Description |
| --- | --- | --- | --- |
| | 0 | 82.88 | Chocolat. Raw sugarcane |
| Var. Tabi | 9 | 83.25 | Raw sugarcane. Floral |
| | 18 | 82.74 | Apple. Caramel |
| | 0 | 53.63 | Stored |
| Var. Castillo General | 9 | 79.50 | Herbal. Cereal |
| | 18 | 52.00 | Stored |
| | 0 | 82.25 | Herbal. Raw sugarcane. Caramel |
| Var. Colombia | 9 | 82.13 | Raw sugarcane. Herbal |
| | 18 | 82.50 | Nuts |

To determine the variations between the attributes according to the SCA protocol, the data obtained for fragrance/aroma, flavor, residual flavor, acidity, body and balance were analyzed (Figure 6).

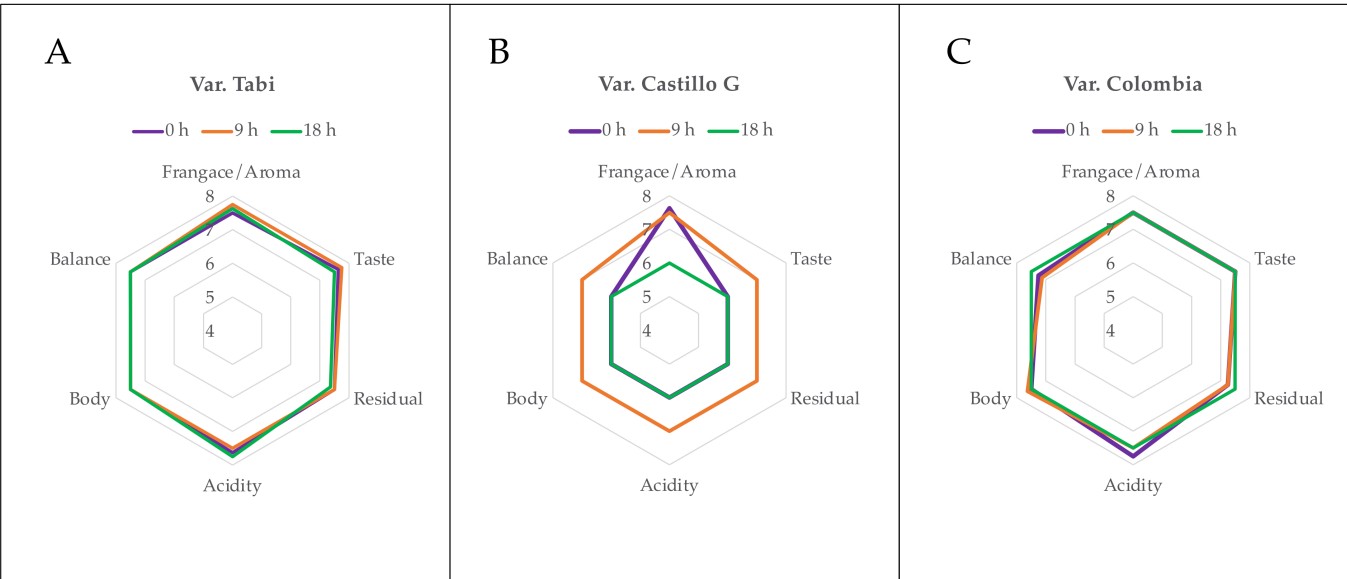

**Figure 6.** Sensory attributes of coffee in three varieties during fermentation (**A**) Var. Tabi. (**B**) Var. Castillo G. (**C**) Var. Colombia.

The fragrance/aroma attribute for the three varieties of coffee during the fermentation process showed average variations of 7.58 and 7.04 at 9 and 18 h, respectively. Higher values (7.75 and 7.63) were observed in the Tabi variety than in the other two varieties. Similarly, the Tabi variety had the highest values for the characteristics of flavor, balance, body, acidity and residual flavor, followed by the Colombia variety. Because a resting defect occurred at the beginning and end of the fermentation period for Var. Castillo General, notable decreases in flavor, residual flavor, acidity, balance and body weight were observed at these times.

## 4. Discussion

The quality of coffee in the cup starts with the variety of coffee planted. All three varieties evaluated in this experiment had the potential to produce high-quality coffee [3]. However, the microbiota naturally present in each of the coffee varieties and on which the fermentation process depends were not known. The molecular diversity approach allowed the identification of general families and genus of microorganisms that were present in the tree varieties at the three fermentation times tested and those which correspond with the ones that were isolated into the pure culture from the mucilage samples. Individual isolates in some cases were identified to species. To ensure that the differences found corresponded to standard processes of the variety, the good practices for the collection and processing of coffee during harvesting, pulping, classification and fermentation were applied [25]. The quality in the cup depends on the state of maturation of the fruits collected [18]. For this reason, the Mediverdes® tool was used to verify that coffee harvesting was excellent. Based on these results, it was possible to guarantee homogeneity in the sample collection process; likewise, with Fermaestro®, it was verified for all three varieties that 18 h of fermentation was sufficient to observe an optimal decrease in the coffee mass, and therefore, the complete degradation of the mucilage [28].

The physicochemical analysis of the mucilage showed that for the three coffee varieties, the pH decreased while the production of organic acids increased with increasing fermentation time. The increase in organic acids was associated with the metabolism of the different microorganisms present during fermentation, and it has been recognized that the presence of these acids can impact the final quality of coffee [9,10]. The concentration and type of organic acid are related to the sensory perception of beverages, such as aroma [46]. The increase in total acidity is due to the increment in lactic acid and acetic acid [10]. Lactic

acid is the result of the fermentation of carbohydrates by bacteria belonging to the lactic acid group through the Embden–Meyerhof–Parnas (homofermentative) pathway or the phosphoketolase pathway (heterofermentative) [47]. The production of acetic acid can be attributed to the presence of AAB that carry out oxidative fermentation of the sugars and the ethanol released by alcoholic fermentation by the yeasts. The action of alcohol and acetaldehyde dehydrogenase enzymes converts alcohol to acetic acid during the fermentation process [48]. Among the three varieties evaluated, Var. Tabi presented the largest population of acetic acid bacteria throughout the process according to molecular analysis, which corresponded with the highest total acidity increase obtained at the end of fermentation. Acetic acid contributes to the bitter taste of coffee, which is usually considered undesirable, but is truly an important taste quality since it participates in the balance of the drink [49]. It should also be noted that in the sensory analysis, this variety was the only one with an apple fruit descriptor that is associated with malic acid, which was not evaluated in this study but may be related to bacteria belonging to the Enterobacteriaceae family that were present throughout the fermentation process. Some genera belonging to Enterobacteriaceae are known to participate in mixed-acid pathway fermentation, giving rise to a mixture of complex organic acids such as lactic, acetic, malic, succinic and formic acids [49]. Therefore, enterobacteria can be considered an important group to examine in future studies due to their negative or positive influence on quality.

Among the microorganisms identified from the coffee fermentation process, various genera and species capable of surviving in a highly acidic environment with limited nutrients stood out. The genera belonging to the LAB had an average initial concentration in the biomass of 6.77 log10 cfu/mL in the three coffee varieties. *Lactobacillus* sp., *Leuconostoc* sp., *Lactococcus* sp. and *Lactiplantibacillus plantarum* are characterized by the production of a large amount of lactic acid as a primary metabolite during their growth. *Leuconostoc* sp. has been found on the surfaces of vegetables, including coffee trees, where the concentration of these bacteria depends on the humidity, sunlight, temperature, and state of maturity of the fruits.

Within the genus *Leuconostoc*, the species *L. mesenteroides* and *L. mesenteroides* ssp. cremoris found in this study are heterofermentative, with relatively low potential for lactic acid production; however, they generate other metabolites of interest during fermentation [17].

Additionally, mixed-acid bacteria, represented by the Enterobacteriaceae family, were found in the fermentation process; in total, eight genera and seven species were found, and these taxa were also reported in Brazil, Australia, Ecuador and China [9,13,16].

Other groups of bacteria found in the coffee fermentation process were the Gram-positive bacterial genera *Micrococcus*, *Staphylococcus* and *Bacillus*, which have the enzymatic capacity to degrade pectin present in the mucilage [50]. Both the LAB group and the *Bacillus* genus, together with acetic bacteria, have been related to characteristics of sensory interest in coffee due to the metabolites generated during their development during fermentation [51].

When studying coffee quality, it is essential to consider water and its management during the pulping, fermentation and washing processes since the microbial load present in water can influence the biochemical process of fermentation and derivatives that impact the sensory quality of coffee [19]. Mesophiles that include microbes from the environment, including those present in water, can develop during fermentation [52]. This group includes the Enterobacteriaceae family, and in this study, *E. coli* and *P. vulgaris* were found, as were *Enterobacter* sp. and *Pantoea* sp.

Yeasts, especially *S. cerevisiae*, in addition to other genera, such as *Pichia* sp. and *Hanseniaspora* sp. are the predominant microorganisms during coffee fermentation, mainly due to the decrease in pH in the fermenting mass and their acidophilic nature [19,52]. Yeasts belonging to the genus Pichia show potential for the development of aroma and flavor. In general, yeasts play a crucial role in the development of sensory characteristics, as stated by Mouret et al. [53]. Most fruit aroma compounds, including esters, are secondary metabolites produced by yeasts during alcohol fermentation [52]; additionally, yeasts have

a biocontrol effect against filamentous fungi that produce mycotoxins. Yeasts also have an enzymatic capacity to degrade various compounds present in the mucilage and can generate volatile and nonvolatile metabolites that are of interest due to their influence on the sensory properties of coffee [51,54].

Bacterial identification at the family and genera levels showed that Var. Castillo General had the greatest diversity at the beginning of the fermentation process with 10 families, of which Enterobacteriaceae and *Acetobacteraceae* were predominant; there is also an assignment of "others", which is similar to *Leuconostocaceae* present in lower abundance; this assignment includes the *Lactobacillaceae* family. At the midpoint in the fermentation process, the number of families was reduced to three, i.e. *Leuconostocaceae*, Enterobacteriaceae and *Acetobacteraceae* but the assignment of "others" is also present. At the end of the fermentation process, these three families were present, but the proportion of *Acetobacteraceae* decreased substantially, with the abundances of *Leuconostocaceae* and *Enterobacteriaceae* being greater. In the case of Var. Colombia, it is interesting that, at the end of the fermentation process, the composition and predominance of the families were very similar to those observed in Var. Castillo General, although at the start and midpoint of the fermentation process, there was a predominance of *Leuconostocaceae* and *Enterobacteriaceae*. In the Tabi variety, the presence of the *Acetobacteraceae* families together with *Leuconostocaceae* and *Enterobacteraceae* was evident from the beginning of the fermentation process; at the midpoint and end of the fermentation process, the abundance of *Enterobacteriaceae* gradually decreased, while the abundances of bacteria belonging to the *Acetobacteraceae* and *Leuconostocaceae* families were maintained. The Tabi variety had the greatest genetic differences, as it originated from a cross between a Timor Hybrid and plants of the Típica and Bourbon varieties.

The changes in the diversity of yeasts and mycelial fungi were similar to those observed in bacteria. The greatest diversity of yeasts and fungi was observed in Var. Castillo General at the initial sampling time, with eight different genera of fungi, followed by Var. Colombia and Var. Tabi. At the midway point of the fermentation process, many genera were still associated with Var. Castillo General, and only at the end of the fermentation process was the number of genera reduced. The three main genera at the end of the fermentation period in the three varieties were the *Saccharomycodaceae* family, *Wickerhamomyces* sp. and *Pichia* sp., corresponding to the same community of microorganisms reported during coffee fermentation in China, Ecuador and Brazil [9,12,55]. The genus *Pichia* sp has been reported as a dominant yeast in coffee fermentation in different countries [20,56,57], resulting in the production of coffees with distinctive flavors for each of them.

Only three genera of filamentous fungi were isolated, *Trichoderma* sp., *Geotrichum* sp. and *Penicillium* sp. The first two genera are reported for the first-time during coffee fermentation, both of which are commonly present in soil. *Trichoderma* sp. and *Penicillium* sp. were identified during the fermentation of all varieties evaluated, and *Geotrichum* sp. was observed only in Var. Tabi and Var. Castillo. The geographical proximity of all coffee crops evaluated could explain the presence of the group of filamentous fungi in the varieties, as reported by Iamanaka et al. [58], i.e. a high incidence of two species of *Penicillium* spp. in coffee in the southeastern region of Sao Pablo, Brazil. Although the presence of filamentous fungi can be associated with risks of mycotoxins in coffee [59], during fermentation, the acidification of the medium and some bacterial metabolites contributed to the decrease in the population of this group, so their role in the fermentation process is not sufficiently understood.

Respect to the observed sugar changes may be directly associated with the microorganisms involved in the fermentation of each variety. In the Tabi and Colombia varieties, decreases in fructose and glucose were also observed, and this loss could be explained by the enzymatic action of the microorganisms present during the fermentation process [10]. In Var. Castillo General, there was an increase in these two sugars, which could also be explained by the enzymatic action of yeasts in the mucilage, attributed to the hydrolysis of sucrose [52]. Notably, at the end of the fermentation period, this variety had the highest levels of fructose and glucose, which are considered the main precursors of volatile com-

pounds during roasting [46]. Therefore, the direct relationship between the sugar contents in the mucilage and seeds should be studied. However, in Var. Castillo General, there was an increase in the sucrose content of the mucilage from the beginning to the midpoint of fermentation, which was not expected and may be associated with a problem during sampling or the accidental entry of fresh plant material that contributed to this uncommon increase in the sucrose content.

At the taxonomic level, the richness and diversity indices allow the characterization of the whole microbial community; in this case, the largest number of bacterial and fungal species was observed at the beginning of fermentation. This was also observed in recent studies in China, where fermentations were evaluated for 36 h and the highest richness and diversity values for both bacteria and fungi were obtained at the beginning of the process [23]. In this study in particular, Var. Castillo General showed a higher alpha diversity than the Colombia and Tabi varieties, according to the ACE and Chao1 indices, for both bacteria and fungi. Notably, the three varieties were obtained from the same farm with the same agronomic management, harvest and processing conditions, which indicated that the variety may have led to differences in the richness of the microbial species. Additionally, among the three varieties, the diversity of microorganisms was the greatest at the beginning of the fermentation process (zero h), and this diversity was reduced at the end of the fermentation process (18 h); similar results were observed by De Oliveira Junqueira et al. [21] after 12 h of coffee fermentation in Colombia and by Cruz-O'Byrne et. al. [24] after 18 h of fermentation. The decrease in the diversity of microbial families may be associated with physicochemical changes in pH, temperature and nutrient availability, which limit the growth of some microorganisms and favor the abundance of LAB and AAB groups.

Genetically, the Castillo General and Colombia varieties are more similar to each other than to other varieties. Both varieties come from crosses between *C. arabica* var. Caturra × Timor Hybrid, a tetraploid population that is used as a rust-resistant parent [3]. Pino et. al. [60] reported the microorganisms found in the rhizosphere of two coffee varieties, Bourbon and Castillo, grown in Popayán-Cauca (Colombia), and they were compared with respect to the organoleptic properties of the coffee cup, demonstrating that each variety of coffee has a distinct microbial profile, which may be related to the plants' physiological, nutritional, and sanitary needs.

In this study, since the three varieties were grown in the same place and under the same conditions, genetic closeness could explain characteristics in the mucilage that gave rise to similarities in the microbial populations of Var. Colombia and Var. Castillo General, as well as differences from what was observed in Var. Tabi. Previously, a rhizosphere study of five species of coffee trees showed that the bacteriomes of *C. arabica* and *C. canephora* are more similar to each other, as *C. arabica* is the result of hybridization *between C. canephora* and *Coffea eugenioides*, which may suggest that *C. arabica* "inherited" the bacteriome from its parent [61].

The cup quality obtained from Var. Tabi and Var. Colombia corresponded to a very good coffee [45]. The highest score (83.25) was obtained across the fermentation period for the coffee obtained from Var. Tabi. In general, Var. Castillo General had the lowest scores due to a moisture defect (i.e. the storage defect), which is not related to the fermentation process. The three varieties have the potential for associations with microorganisms that allow excellent quality coffee to be obtained. To achieve SCA quality, beverages need to come from specialty coffee with no defects and at least 80 points on the scale for specialty coffee [45]. Furthermore, unlike Var. Castillo General and Var. Colombia, in Var. Tabi, in addition to the presence of LAB and AAB belonging to the genera *Gluconobacter*, showed *Acetobacter*, *Frauteria* and yeast belonging to *Pichia* were present throughout the fermentation process; therefore, the high quality obtained and the sensory characteristics, such as the chocolate, floral and apple flavors, may be due to the influence of all of these microorganisms. Although the agronomic practices and the agroforestry system of the crop have been shown to have an impact on the chemical and quality profiles

of coffee [62], importantly, the geographical conditions and agronomic management of the coffee crops evaluated were the same, and the results of this study suggest that the microbial communities during fermentation may be different according to the variety; similar results were found by other authors [60,63]. However, the differences between the microbial communities in fermentation also allowed us to establish that there is no single way to obtain high-quality washed coffee; rather, the biochemical changes caused by microorganisms during fermentation and the coffee beans can enhance the intrinsic quality of each variety.

Although the microbial groups associated with the Castillo General and Colombia varieties were similar, the differences relative to Var. Tabi included the predominance of microbes belonging to the AAB group in Var. Tabi. In contrast, Var. Colombia and Var. Castillo General were mainly associated with LAB belonging to the family *Lactobacillaceae* and genera *Leuconostoc*, *Enterobacteria*, *Weissella* sp. and *Tatumella* sp. and the yeast *Wickerhamomyces*, which account for the high quality obtained and the sensory characteristics of raw sugarcane and, herbal and nutty notes. All of the interactions of the main microorganisms found to be involved in the metabolism of sugars give rise to the formation of pyrazines involved in the Maillard reaction during the roasting of coffee beans [46]. In these results, 2,6-dimethylpyrazine, 2-ethyl-3-methyl-pyrazine, and 2-ethyl-2,5-dimethyl-pyrazine can be correlated with the sweet, nut, caramel, roasted, and chocolate notes detected in the different coffee samples [54]. Future studies should be directed not only to identify the production of organic compounds in each microorganism involved in the fermentation of coffee, but also to evaluate the complex microbial interactions that could be influenced by the coffee variety.

## 5. Conclusions

This study is the first to accurately describe the microbiological characteristics of fermentation in three coffee varieties from Colombia, providing evidence of the differences in the dominant microbial groups during the fermentation depending on the variety. The three varieties showed the ability to produce specialty coffees if they were processed according to good practices. The results indicated that native microbial communities occur naturally during coffee processing. Fermentation depends on a microbial consortium of mixed-acid bacteria (Enterobacteriaceae, *Tatumella* sp.), lactic acid bacteria (*Leuconostoc* sp. *Weissella* sp. and *Lactobacillaceae*), acetic acid bacteria (*Gluconobacter* sp. and *Acetobacter* sp.) and fungi *Saccharomycodaceae*, and *Pichia* rather than on a single microorganism, the variety of coffee, and the environmental characteristics of the area where wet coffee processing and fermentation take place. Finally, all of these factors affect the development of microbial communities during fermentation. This research establishes new perspectives for the improvement of fermentation processes, and cup quality of Colombian coffee.

**Supplementary Materials:** The following supporting information can be downloaded at: https://www.mdpi.com/article/10.3390/agriculture13101980/s1, Figure S1: Rarefaction analysis of 16S rDNA sequencing; Figure S2: Rarefaction analysis of ITS sequencing. Table S1: Characterization of the coffee harvest title; Table S2: Results of deep sequencing of 16S rDNA; Table S3: Results of deep sequencing of ITS.

**Author Contributions:** Conceptualization, C.E.G., L.H.-S. and B.P.-C.; methodology, L.H.-S., A.O., B.P.-C. and R.P.-S.; validation, C.E.G., L.H.-S. and L.N.-E.; formal analysis, L.H.-S., B.P.-C., A.O., L.N.-E. and R.P.-S.; writing—original draft preparation, C.E.G. and L.H.-S.; Data Curation, L.N.-E.; writing—review and editing, C.E.G., L.H.-S., L.N.-E., B.P.-C. and R.P.-S.; supervision, C.E.G. All authors have read and agreed to the published version of the manuscript.

**Funding:** This research was funded by Contract No. 2019 02 1486 for the development of scientific and technological activities, signed between the Department of Cesar and the National Federation of Coffee Growers of Colombia for the purpose of "Experimental Development for the improvement of the Competitiveness of the coffee sector of the Department of Cesar", code BPIN 2017000100036,

financed by the Fund for Science, Technology and Innovation-FCTeI, of the General System of Royalties—SGR and National Coffee Research Center (Cenicafé) project number POS101024.

**Institutional Review Board Statement:** Sampling was carried out under the guidelines of the Collection Framework Permit 01749 granted by the Environmental Licenses Authority-ANLA to the Universidad de Santander, Colombia.

**Informed Consent Statement:** Not applicable.

**Data Availability Statement:** The datasets presented in this study can be found in online repositories. The names of the repositories and accession number(s) can be found at https://www.ncbi.nlm.nih.gov/Bioproject: PRJNA940996 (accessed on 1 May 2023).

**Acknowledgments:** We thank personnel from the Pueblo Bello station in Cesar, José Enrique Baute Belarcazar, Pedro Hernandez, personnel of the Cesar-Guajira Committee of Colombian Coffee Growers Federation (FNC), Helvert Acuña, Liseth Javela and field promotors for logistical support. We also thank the Multilab laboratory where the analysis of minor elements was carried out, Juan F Alzate and Katherine Bedoya from the National Center for Genomic Sequencing, University of Antioquia, and the FASPLAN Company for discussions about the metataxonomic analysis.

**Conflicts of Interest:** The authors declare no conflict of interest.

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
