# Peer review of "Physical–Chemical and Metataxonomic Characterization of the Microbial Communities Present during the Fermentation of Three Varieties of Coffee from Colombia and Their Sensory Qualities"

_agriculture, doi:10.3390/agriculture13101980_

Round 1
Reviewer 1 Report
The authors studied physical-chemical properties and microbial communities during the fermentation process of Coffea arabica L varieties. Multiple physico-chemical parameters (temperature, pH, total acidity) as well as sugars (sucrose, glucose and fructose) and protein content were measured during the fermentation. Moreover number of CFUs belonging different bacterial groups (including lactic acid bacteria and acetic acid bacteria) and fungi (yeasts and mycelial fungi) were calculated, some pure cultures were isolated. Microbial communities structure were studied using high throughput sequencing of V3V4 region of 16S rRNA gene (for bacteria) and ITS (for fungi). This work is quite interesting as well as most of the results are well presented but several points should be improved:
-Аuthors provided amounts of lactate and acetate but they measured only total acidity but not individual organic acids. This situation is completely unclear – how authors calculated amounts of individual organic acids? In the same time concentrations of organic acids is important part of the study and many points of view in Discussion are associated with these data. Thus this part must be improved and clarified.
-Аuthors estimated numbers of CFUs belonging to different groups of bacteria and fungi in the samples. However the data looks quite strange because according to the data there was no growth of microorganisms during fermentation. For example, numbers of CFUs of LAB, one of the main functional group during fermentation, in Var. Castillo General were ~5.89 millions (Log10=6.77) at zero hours of fermentation and ~2 millions (Log10=6.30) at final stage of fermentation. There are quite similar situations in other samples and among other microbial groups. Also there are mistakes in Total microbial load calculation.
-Аim of isolation of pure cultures of bacteria and fungi is unclear. For studying of microbial community authors used Illumina sequencing of marker gene fragments (V3V4 region of 16S rRNA and ITS for bacteria and fungi, respectively), so why was it necessary to isolate pure cultures? The comparison of results, obtained using cultural and molecular approaches, in Discussion is absent.
-Discussion section is massive and difficult to read. There many duplicates from Results and several from Introduction and Methods. This section should be rewritten and improved
-Conclusions contain some general thoughts weakly related to results of the study (see comments L797-700, L800) and it must be fixed. Moreover summary about microbial communities studied in this work should be present.
Minor comments
L20: please add “…microorganisms isolated into pure cultures”
L227: Sections 2.4.1 and 2.4.2 should be shortened
LL354-355: why authors provide percentage of green fruits less than 0.38%? According to data in Supplementary Table S1 the percentages is 0.49% for Tabi var. (1/205) and Castillo General var. (1/203). Or is there specific calculation method (if yes, please explain it in footnotes in Supplementary Table 1)?
LL367-369: “For Var. Castillo General and Var. Tabi, respective values of 5.00 and 4.00° Brix were observed. For these varieties, similar values were observed in the initial and middle periods….”
According to Table 1 Brix value for Var. Tabi decreased from 4.0° to 2.0° during first 9 hours of fermentation, so these values don't seem to be similar.
LL374-379: How authors calculate proportions of acetate and lactate if they measure only total acidity? There is formula in Methods section but actually I still do not understand how concentration of individual organic acids can be estimated without direct measurements (e.g. using HPLC).
L428: There are quite strange data in the Table 3: almost all microbial groups present in similar CFU numbers at initial and final stages of the fermentation in all three coffee varieties. Microorganisms cultures should show typical curve during growth with exponential increasing of cell number.
Moreover, Total microbial loads were calculated incorrectly because authors just summarized Log10 values and consequently obtained the final values for multiplication of arguments, but not sum (loga x + loga y = loga xy). Please correct it.
Numbers of CFUs for acetic acid bacteria are absent in Table3.
L455: Please replace “penicillium sp.” with “Penicillium sp.” in the Table 5
L478: Are points from the one coffee variety on the Figure 1 indicate samples with different time of fermentation? If yes please indicate this info on the Figure 1. There is similar question about Supplementary Figure 3. And probably it will be useful to combine Figure 1 and Figure S3 as Figure 1A and 1B.
LL490-542: Please rewrite taxa names in italics. And please add relative abundances values for key taxa (bacterial families level, bacterial genera level, fungi families level, fungi genera level) into the text.
L498: There is no Figure 3 in the current version of manuscript. Please add it.
LL575-588: This doesn't sound like a discussion (rather there is duplicate of Introduction and Methods section).
LL595-596: see comment for LL374-379
LL599-600: see comment for LL374-379
L639: How authors explain that they isolated strains of Latilactobacillus and Lactococcus but these genera were not detected using sequencing of V3V4-regions of 16S rRNA gene?
L654: “…reported in recent research…” recent research of what? Please specify it.
LL797-799: “The three varieties showed the ability to produce specialty coffees if they were processed according to the practices established by the National Federation of Coffee Growers”.
Actually this point is not directly supported by the results.
L800: “The quality is influenced by spontaneous fermentation processes”
According to Figure S4 sensory attributes (SCA scores) changed very little during the fermentation for Var. Tabi and Var. Colombia.
L801: What native microbial communities? Please specify main bacterial and fungal components.
Author Response
Agriculture-2629028
The authors would like to thank the reviewer N°1 for all the comments that helped us to improve the quality and clarity of the manuscript. In the new version of the paper all deletion are crossed out, the addition the specific changes suggested by this reviewer are in red color other changes aye highlight in yellow or green
Each one of the comments were answering.
Reviewer1
-Аuthors provided amounts of lactate and acetate but they measured only total acidity but not individual organic acids. This situation is completely unclear – how authors calculated amounts of individual organic acids? In the same time concentrations of organic acids is important part of the study and many points of view in Discussion are associated with these data. Thus this part must be improved and clarified.
LL374-379: How authors calculate proportions of acetate and lactate if they measure only total acidity? There is formula in Methods section but actually I still do not understand how concentration of individual organic acids can be estimated without direct measurements (e.g. using HPLC).
Authors reply
The determination of organic acids was carried out based on the methodology proposed by Mindreau Ganoza et al., (2016). The result corresponds to an indirect estimation of the organic acid.
The reference [31] was included in the equation (line 196) and references.
Reference
- Mindreau Ganoza, E.; Juscamaita Morales, J.; Williams León De Castro, M. Estabilización de heces humanas provenientes de baños secos por un proceso de fermentación ácido láctica. Ecol. Apl. 2016, 15, 143, doi:10.21704/rea.v15i2.754.
Reviewer1
-Аuthors estimated numbers of CFUs belonging to different groups of bacteria and fungi in the samples. However the data looks quite strange because according to the data there was no growth of microorganisms during fermentation. For example, numbers of CFUs of LAB, one of the main functional group during fermentation, in Var. Castillo General were ~5.89 millions (Log10=6.77) at zero hours of fermentation and ~2 millions (Log10=6.30) at final stage of fermentation. There are quite similar situations in other samples and among other microbial groups. Also there are mistakes in Total microbial load calculation.
Authors reply
Table 3 (lines 435-436) was reviewed in deep. The column related with the total number of microorganisms was deleted, because it was confusing and did not provided extra information. Some corrections were done in two data that are identified in red -yellow in the table.
It is important to point out that the microorganism count was done in a specific point-time of fermentation (0h, 9h and 18h) and the number and proportion of the different microorganism will change depending on the media and environmental conditions. It is because of that, that the numbers of the different groups can increase or decrease as is observed in the table (3) and also in Figure 2-3 and Figure 4-5
-Аim of isolation of pure cultures of bacteria and fungi is unclear. For studying of microbial community authors used Illumina sequencing of marker gene fragments (V3V4 region of 16S rRNA and ITS for bacteria and fungi, respectively), so why was it necessary to isolate pure cultures? The comparison of results, obtained using cultural and molecular approaches, in Discussion is absent.
Authors reply
We are interested in understanding and in the future to be able to manipulate the fermentation process in coffee. We want to use the microorganisms not only yeast but also bacteria to improve the coffee quality, but before doing this we need to know what are the microorganisms and isolate them, and understand their relationships and be able to growth them.
Respect to the comparison of results, obtained using cultural and molecular approaches. The molecular approach gives us the general Families and genus of microorganisms that were present in the tree varieties at the three fermentation times tested and those correspond with the ones that we found when we isolate the microorganisms from the mucilage samples. Individual isolates allowed to identified some species. This information was added in the discussion line 638-641
.
- Reviewer1
Discussion section is massive and difficult to read. There many duplicates from Results and several from Introduction and Methods. This section should be rewritten and improved
Authors reply
Base on your suggestion we short the discussion, making it more specific. We reorganized the discussion moving some of the paragraph the one that were moved are in green color.
We crossed out in the discussion the repeated or general information.
The discussion was focus first in shows how the fermentation induces the production of acids, what microorganisms base on that environment we could isolate. Then we show that the molecular approach corroborate the information about microorganisms Family and Genus and allow us to understand the different population in the coffee varieties and in the different fermentation time and how that can be correlated with the coffee quality.
- Reviewer1
-Conclusions contain some general thoughts weakly related to results of the study (see comments L797-700, L800) and it must be fixed. Moreover summary about microbial communities studied in this work should be present.
LL797-799: “The three varieties showed the ability to produce specialty coffees if they were processed according to the practices established by the National Federation of Coffee Growers”.
Actually this point is not directly supported by the results.
Authors reply
The Reviewer is right. Part of information in the conclusion was deleted in order to make it more specific and accurate according with the results.
New conclusion:
This study is the first to accurately describe the microbiological characteristics of fermentation in three coffee varieties from Colombia, providing evidence of the differences in the dominant microbial groups during the fermentation process, depending on the variety. The results indicated that native microbial communities occur naturally during coffee processing. Fermentation depends on a microbial consortium rather than on a single microorganism, the variety of coffee, and the environmental characteristics of the area where wet coffee processing and fermentation take place. This research establishes new perspectives for the improvement of fermentation processes, sensory properties and cup quality of Colombian coffee.
Reviewer 1
Minor comments
- L20: please add “…microorganisms isolated into pure cultures”
Authors reply
It was done
- L227: Sections 2.4.1 and 2.4.2 should be shortened
Authors reply
We only refer to general information for microorganism’s isolation.
- LL354-355: why authors provide percentage of green fruits less than 0.38%? According to data in Supplementary Table S1 the percentages is 0.49% for Tabi var. (1/205) and Castillo General var. (1/203). Or is there specific calculation method (if yes, please explain it in footnotes in Supplementary Table 1)?
Authors reply
The percentage is calculated according to standardized and validated reference values during the development of the tool and the information was added as a food note in supplementary table 1.
*According color chart Cromacafé (Peñuela-Martínez, Guerrero and Sanz-Uribe, 2022)
Peñuela-Martínez, A. E., Guerrero, Á., & Sanz-Uribe, J. R. (2022). Cromacafé® Herramienta para identificar los estados de madurez de las variedades de café de fruto rojo. Avances Técnicos Cenicafé, 535, 1-8. https://doi.org/10.38141/10779/0535
(Guerrero, Á. et al., 2022). Guerrero, Á.; Sanz-Uribe, J.R.; Peñuela-Martínez, A.E.; Ramírez, C.A. Mediverdes®: Un Método Para Medir La Calidad de La Recolección Del Café En El Campo. Av. Téc. Cenicafé 2022, 536, 1–8, https://doi.org/10.38141/10779/0536
- LL367-369: “For Var. Castillo General and Var. Tabi, respective values of 5.00 and 4.00° Brix were observed. For these varieties, similar values were observed in the initial and middle periods….”
- According to Table 1 Brix value for Var. Tabi decreased from 4.0° to 2.0° during first 9 hours of fermentation, so these values don't seem to be similar.
Authors reply
The reviewer is right, an error was made in the writing. The description of the result was corrected on lines 370-372.
- L428: There are quite strange data in the Table 3: almost all microbial groups present in similar CFU numbers at initial and final stages of the fermentation in all three coffee varieties. Microorganisms cultures should show typical curve during growth with exponential increasing of cell number.
Authors reply
We are not counting the growth of the microorganisms we are only counting the total number in a specific point time ( 0, - 9 – 18h) it is not a growth curve. In some cases, because after 9 o 18 hour the pH change or the temperature, the total number do not increase, even can decrease.
- Moreover, Total microbial loads were calculated incorrectly because authors just summarized Log10 values and consequently obtained the final values for multiplication of arguments, but not sum (logax + loga y = loga xy). Please correct it.
- Numbers of CFUs for acetic acid bacteria are absent in Table3.
Authors reply
Table 3 (lines 435-436) was reviewed in deep. The column related with the total number of microorganisms was deleted, because it was confusing and did not provided extra information. Some corrections were done in two data that are identified in red -yellow.
The acetic acid bacteria were not counted and the isolation was kind of difficult
- L455: Please replace “penicillium sp.” with “Penicillium” in the Table 5
Authors reply
The reviewer is right, the change was done. Table 5, line 465-466
- L478: Are points from the one coffee variety on the Figure 1 indicate samples with different time of fermentation? If yes please indicate this info on the Figure 1. There is similar question about Supplementary Figure 3. And probably it will be useful to combine Figure 1 and Figure S3 as Figure 1A and 1B.
Authors reply
The reviewer is right, the change was done.
We combine Figure1 and Figure from the supplementary material 3, in a new figure 1.
Figure 1. NMDS analysis of A. Bacteria and B. Fungi associated with the three varieties evaluated Lines 505-507.
- LL490-542: Please rewrite taxa names in italics. And please add relative abundances values for key taxa (bacterial families level, bacterial genera level, fungi families level, fungi genera level) into the text.
Authors reply
All family and genus names were check out and rewrite in italics. Following the recommendation of the reviewer 3, the sp is not in italic.
- L498: There is no Figure 3 in the current version of manuscript. Please add it.
Authors reply
Figure 3 was added line 543 to 556
- LL575-588: This doesn't sound like a discussion (rather there is duplicate of Introduction and Methods section).
Authors reply
We change part of the discussion at the beginning, deleting information that was in the introduction lines 648-651. In addition we add part of the information that you suggested in line 638-641
- LL595-596: see comment for LL374-379
Authors reply
The determination of organic acids was carried out based on the methodology proposed by Mindreau Ganoza et al., (2016). The result corresponds to an indirect estimation of the organic acid.
The reference [31] was included in the equation (line 175)
Reference
- Mindreau Ganoza, E.; Juscamaita Morales, J.; Williams León De Castro, M. Estabilización de heces humanas provenientes de baños secos por un proceso de fermentación ácido láctica. Ecol. Apl. 2016, 15, 143, doi:10.21704/rea.v15i2.754.
- L639: How authors explain that they isolated strains of Latilactobacillus andLactococcus but these genera were not detected using sequencing of V3V4-regions of 16S rRNA gene?
Authors reply
The sequencing of V3V4-regions of 16S rRNA gene identified a group of microorganisms at level of family and genus call “ other” it is possible that both Lactilactobacillus and Lactococcus be classified in that group,
- L654: “…reported in recent research…” recent research of what? Please specify it.
Authors reply
Author is right the word is deleted. Line 704
- L800: “The quality is influenced by spontaneous fermentation processes”
Authors reply
Taking this recommendation into account, the sentence was deleted (line875). The conclusion was re-organized.
- According to Figure S4 sensory attributes (SCA scores) changed very little during the fermentation for Var. Tabi and Var. Colombia.
Authors reply
Author is right. However, At the suggestion of another reviewer Figure S4 was included in the main manuscript (Figure 6. Line 624).
- L801: What native microbial communities? Please specify main bacterial and fungal components.
Authors reply
Line 879- The native microbial communities correspond to microorganisms present in the environment that can be closely related to specific coffee varieties and the place where the coffee is growth that develop during fermentation, which in turn is not a single microorganism but a microbial consortium"

Reviewer 2 Report
A. Title:
a. Physical-chemical and metataxonomic characterization of the 2 microbial communities present during the fermentation of 3 three varieties of coffee from Colombia and their sensory qualities
B. Abstract:
a. The results are not efficiently summarized in the abstract kindly improve it
C. Introduction:
a. Mention at least some of the latest references related to microbial communities present fermentation PROCESS
b. The introduction is over generalized. Improve it
D. Materials and Methods:
a. Italicize the names of microorganisms.
E. Discussion
a. Please add some latest references and improve the discussion further.
b. Seems over generalized.
F. Plagiarism : Reduce the plagiarism up to 15%
G. References should be journal pattern
H. Conclusions must be shorter
Nill
Author Response
Agriculture-2629028
The authors would like to thank the reviewer N°2 for all the comments that helped us to improve the quality and clarity of the manuscript. In the new version of the paper all deletions are crossed out), the addition and changes specifically suggested by this reviewer are highlight in green color.
Each one of the comments were answered.
Reviewer2
- Abstract:
- The results are not efficiently summarized in the abstract kindly improve it
Authors reply
We add some information in the abstract (line 20), we reorganize the abstract, first we put all the information about the microorganisms and then the information about the quality, in such a way that the results are easily understood. The changes are indicated in green color.
- Introduction:
- Mention at least some of the latest references related to microbial community’s present fermentation PROCESS
- The introduction is over generalized. Improve it
Authors reply
Two new references corresponding to new publications were added to the paper and the information is in lines 73 and Line 87-88
Reference 18. Apolo, D.; Fernández, J.M.; Benítez, Á.; Figueroa, J.G.; Estrada, K.; Cruz, D. Phenotypic and Molecular Characterization of Yeast Diversity Associated to Postharvest Fermentation Process of Coffee Fruits in Southern Ecuador. Diversity 2023, 15, 984, doi:10.3390/d15090984. (Coffee in Ecuador and var. Geisha characteristics)
Reference 23. Shen, X.; Wang, B.; Zi, C.; Huang, L.; Wang, Q.; Zhou, C.; Wen, W.; Liu, K.; Yuan, W.; Li, X. Interaction and Metabolic Function of Microbiota during the Washed Processing of Coffea Arabica. Molecules 2023, 28, 6092, doi:10.3390/molecules28166092. (Coffee in Yunna China where it was evident that the metabolic functions of microorganisms can vary according to the interaction with other microorganisms in the environment to obtain washed coffees.
We change part of the introduction to make it more specific in Colombian coffee, the region in which the coffee fermentation took place and the coffee varieties that were tested. This new information allowed it to be more specific decreasing the sense of generalizing.
- Materials and Methods:
- Italicize the names of microorganisms.
Authors reply
We check all of them and we did italicize them.
- Discussion
- Please add some latest references and improve the discussion further.
- Seems over generalized.
Authors reply
We add the reference 23, this allow us to explain in a better way the richness and diversity indices, that are comparable to the one reported in the reference. Lines 798-801.
We reduce the discussion and change part of the order of the discussion to make it more specific and clear. Paragraphs in green were changed. To make more specific with our results we change information in lines 655 to 660.
To reduce over generalization in discussion: all crossed lines were deleted.
- Plagiarism :Reduce the plagiarism up to 15%
Authors reply
We check plagiarisms with the programs turniting. The coincidence was 8%
- Referencesshould be journal pattern
Authors reply
All the references were check and corrected
- Conclusions must be shorter
Authors reply
We reduce the conclusion. Lines 870-872-874-875-876 were deleted. The conclusion was re-write

Reviewer 3 Report
Dear Authors,
The manuscript entitled "Physical-chemical and metataxonomic characterization of the microbial communities present during the fermentation of three varieties of coffee from Colombia and their sensory qualities", is a summary of a very interesting and necessary work to understand the complex interactions that take place in the fermentation of coffee beans. The work is therefore relevant and adds to the knowledge in this area.
Despite the merits of the work, it is necessary to amend some minor typing errors and other omissions that I believe would improve the quality of the work and the contribution that this study could make to those interested in this field of coffee fermentation.
These points are:
1. I consider that some of the information that is placed in the "supplementary materials" should be, due to its relevance, within the main text.
2. Figure 3 is omitted.
3. Some values shown should be thoroughly revised.
4. The discussion, although it generally addresses the relevance of the work, does not explain well the differences observed among the microorganisms isolated from the three strains, nor does it discuss the possible causes of such differences.
5. Other minor remarks are pointed out directly in the original manuscript attached.
For all these reasons, and despite the merits of the work, I believe that it should be thoroughly checked, and the above-mentioned points should be corrected before being published in the journal.
Congratulations on the work done.
Regards,
Reviewer

Author Response
Agriculture-2629028
The authors would like to thank the reviewer N°3 for all the comments that helped us to improve the quality and clarity of the manuscript. In the new version of the paper all the addition or changes are highlight in yellow and the deletion are crossed out and highlight in yellow color.
Each one of the comments were accepted and answering. We also review the changes that were directly done in the manuscript.
Reviewer 3
- I consider that some of the information that is placed in the "supplementary materials" should be, due to its relevance, within the main text.
Authors reply
Reviewer is right. Base on the comments from the reviewer and the remarks that were pointed out directly in the original manuscript. We moved the following tables and figures from supplementary material to the main text.
- Table 4 that was in the supplementary material is in the main text as Table 6 (Table 6. Bacterial diversity during the fermentation of three coffee varieties) lines 477-480.
- Table 5 that was in the supplementary material is in the main text as Table 7 (Table 7. Fungal diversity during the fermentation of three coffee varieties) lines 570-572.
- Figure 4 that was in in the supplementary material is in the main text as Figure (Figure 6. Sensory attributes of coffee in three varieties during fermentation A) Var. Tabi. B) Var. Castillo G. C) Var. Colombia) lines 620-623.
The supplementary file was modified due to these changes.
Reviewer 3
- Figure 3 is omitted.
Authors reply
Reviewer is right. We add figure 3 in lines 544-559.
Reviewer 3
- Some values shown should be thoroughly revised.
Authors reply
We reviewed all values in table 1 and table 2, as indicated in the remarks that were pointed out directly in the original manuscript by the reviewer.
Reviewer 3
- The discussion, although it generally addresses the relevance of the work, does not explain well the differences observed among the microorganisms isolated from the three strains, nor does it discuss the possible causes of such differences.
Authors reply
In line 737-753 the differences at level of Bacteria are explained.
In lines 755 to 765 differences at level of fungi are explained
In lines 815-829, we explain also, since the three varieties were grown in the same place and under the same conditions, genetic closeness could explain characteristics in the mucilage that gave rise to similarities in the microbial populations of
- Colombia and Var. Castillo General, as well as differences from what was observed in Var. Tabi.
Reviewer 3
- Other minor remarks are pointed out directly in the original manuscript attached.
Authors reply
We corrected all the remarks done by the reviewer and indicate the changes in yellow.
- Table 1. Separation in the + standard error was done (Lines 375-376).
In addition, the discussion about the differences in % of lactic acid and acetic acid from the beginning to the end was added as suggested by the reviewer in the document.
Line 377-381. In all cases, an increase in the % values of organic acids were evident from the beginning to the end of fermentation, with lactic acid always being the highest proportion with percentages of 12.6, 10.0 and 8.2%, while in acetic acid values of 8.4, 6.7 and 5.4% in the vars. Tabi, General Castillo and Colombia respectively.
- Table 2. Separation in the + standard error was done and the data in yellow was corrected.
two data error were corrected (values from sucrose and glucose in mucilage from the Var. Tabi). Lines 417-418.
- Table 3 was also corrected- a line at 0 time was erase. Line 435-436.
- Table 4 and 5 was delimited with a vertical line in order to separate the different group of microorganisms. Lines 445-446 and lines 446-447.
- The word hours was changed to “h” throughout the document. Likewise, minutes to “min”
- Line 192: the word alkaline sodium hydroxide was removed
- Line 198: In the formula the units (ml) were specified.
- The scientific names that were not in italic style were corrected and the sp ending was also corrected as well as spp.

Round 2
Reviewer 1 Report
Some parts of the manuscript still need to be improved:
Authors reply
The determination of organic acids was carried out based on the methodology proposed by Mindreau Ganoza et al., (2016). The result corresponds to an indirect estimation of the organic acid. The reference [31] was included in the equation (line 196) and references.
Reference
31. Mindreau Ganoza, E.; Juscamaita Morales, J.; Williams León De Castro, M. Estabilización de heces humanas provenientes de baños secos por un proceso de fermentación ácido láctica. Ecol. Apl. 2016, 15, 143, doi:10.21704/rea.v15i2.754.
-I checked this reference and it does not contain a description of the method but just citation of “Peralta Verán R.L. 2010. Determinación de parámetros óptimos en la producción de fast biol usando las excretas del ganado lechero del establo de la UNALM”. Firstly, I cannot find Peralta, 2010 in public sources. Secondly, it seems that this method could be used in cases when samples contain single organic acid (e.g. only lactate amount was estimated in paper Mindreau Ganoza et al.) but not several organic acid (as in current paper). Thus, authors must 1) either provide the reference (preferably in English) in which method of calculation of individual organic acid proportion in mix (without direct measurement) is described or 2) remove data about acetate and lactate concentrations and use only total acidity values.
Authors reply
Table 3 (lines 435-436) was reviewed in deep. The column related with the total number of microorganisms was deleted, because it was confusing and did not provided extra information. Some corrections were done in two data that are identified in red -yellow in the table. It is important to point out that the microorganism count was done in a specific point-time of fermentation (0h, 9h and 18h) and the number and proportion of the different microorganism will change depending on the media and environmental conditions. It is because of that, that the numbers of the different groups can increase or decrease as is observed in the table (3) and also in Figure 2-3 and Figure 4-5
-I do not understand authors' answer. For example, let’s take CFUs number for Lactic acid bacteria in Var. Tabi: at initial point (0h) there were ~ 6025595 of CFUs, while at middle point (9h) and end of fermentation (18h) – 2818382 and 38018 of CFUs, respectively. In all these points, media and environmental conditions which used for cultivation were same so changes in CFUs number should caused by microbial growth. However, according to the data highest number of CFUs (which should be approximately equal to the number of cells) was observed before start of fermentation. There are many such cases in the data. If bacteria did not grow during fermentation, how did they digest coffee fruit mucilage and produce organic acids or other metabolites?
Authors reply
The reviewer is right, the change was done. We combine Figure1 and Figure from the supplementary material 3, in a new figure 1.
Figure 1. NMDS analysis of A. Bacteria and B. Fungi associated with the three varieties evaluated
-But still there is no indication how dots on NMDS plot correspond to time points. Please add this info on the Figure 1.
Authors reply
The sequencing of V3V4-regions of 16S rRNA gene identified a group of microorganisms at level of family and genus call “ other” it is possible that both Lactilactobacillus and Lactococcus be classified in that group,
-Please check it. And if Lactilactobacillus and Lactococcus classified as “Other”, specify it in Results and Discussion sections.
-Moreover, relative abundance for key taxa (bacterial families level, bacterial genera level, fungi families level, fungi genera level) should be indicated within the text.
Authors reply
Line 879- The native microbial communities correspond to microorganisms present in the environment that can be closely related to specific coffee varieties and the place where the coffee is growth that develop during fermentation, which in turn is not a single microorganism but a microbial consortium"
-I know what “native microbial community” means. The question to the authors was about specific key components of communities studied in the work, because coffee microbiomes composition is main focus of the study, the some summary about it should be added to Conclusions section.
Author Response
Dear Reviewer
We want to thanks Reviewer 1 again for reviewer the re-submission of the paper. We again check and answer each one of the comments and we hope to be able to answer to satisfaction your inquiries.
Base on the comments from you and reviewer 3, that accept some of the changes in the first round of reviewing. We proceed to accept in the document the changes related with the first round of reviewing.
Base on your new comments, in the R2-version of the paper all the additions are identified in red color. The deletions are crossed out, some remarks are highlighted in yellow color.
____________________________________________________________________
1.Reviewer
The determination of organic acids was carried out based on the methodology proposed by Mindreau Ganoza et al., (2016). The result corresponds to an indirect estimation of the organic acid. The reference [31] was included in the equation (line 196) and references.
Reference
- Mindreau Ganoza, E.; Juscamaita Morales, J.; Williams León De Castro, M. Estabilización de heces humanas provenientes de baños secos por un proceso de fermentación ácido láctica. Ecol. Apl. 2016, 15, 143, doi:10.21704/rea.v15i2.754.
-I checked this reference and it does not contain a description of the method but just citation of “Peralta Verán R.L. 2010. Determinación de parámetros óptimos en la producción de fast biol usando las excretas del ganado lechero del establo de la UNALM”. Firstly, I cannot find Peralta, 2010 in public sources. Secondly, it seems that this method could be used in cases when samples contain single organic acid (e.g. only lactate amount was estimated in paper Mindreau Ganoza et al.) but not several organic acid (as in current paper). Thus, authors must 1) either provide the reference (preferably in English) in which method of calculation of individual organic acid proportion in mix (without direct measurement) is described or 2) remove data about acetate and lactate concentrations and use only total acidity values.
- Authors Answer
We could not find a reference in English that explain this indirect method for calculating the two acids. In addition, we check with our chemists, and the reviewer is right, the calculation is not completely appropriate (it can be done only for lactic acid in some foods: milk and beer). We decide to remove the information from the manuscript methodology Lines 194-198. The reference [31] was also change to a new one, for the evaluation of Total Acidity (line 194 and the new reference is in line 991).
Also, the data in the results table 1, Line 374-375 was removed.
We only leave in table 1, total acidity values.
The result discussion in lines 376-383 will be deleted
The organic acid contents in the mucilage (Table 1) increased during the fermentation process for the three varieties evaluated. The increment was higher in lactic acid compare to acetic acid. In the case of lactic acid content, the biggest increase from the beginning to the end of the fermentation process was observed in Var. Tabi where a 12.6 % of increment was observed, follow by Var. Castillo with a 10%. In Var. Colombia the increment was the lowest (8.22 %). For acetic acid, an increase from the beginning to the end was also observed. It was the highest in Var. Tabi (8.41%), follow by Var. Castillo (6.74%) and Var. Colombia was the one with lower differences (5.5%).
They will be changes for this one:
The organic acid contents in the mucilage (Table 1) increased during the fermentation process for the three varieties evaluated. The highest total acidity at the beginning of the process was observed in var. Tabi. This one was also the one that showed the highest total acidity increase from T1 to T3 (the increment was 700 mg/L). Followed by Var. Castillo that showed an increase of 560 mg/l from the beginning to the end of the fermentation process. Var Colombia was the one with the lowest differences in total acidity, with 124 mg/L increment from the beginning to the end of the fermentation process (lines 385-392)
Other changes related to this topic were done.
Line 668. 680. 681 Information was added
Lines 669-670 will be deleted together with line 674, 681.
- Reviewer
-I do not understand authors' answer. For example, let’s take CFUs number for Lactic acid bacteria in Var. Tabi: at initial point (0h) there were ~ 6025595 of CFUs, while at middle point (9h) and end of fermentation (18h) – 2818382 and 38018 of CFUs, respectively. In all these points, media and environmental conditions which used for cultivation were same so changes in CFUs number should caused by microbial growth. However, according to the data highest number of CFUs (which should be approximately equal to the number of cells) was observed before start of fermentation. There are many such cases in the data.
If bacteria did not grow during fermentation, how did they digest coffee fruit mucilage and produce organic acids or other metabolites?
- Authors Answer
There are other bibliographic references that shows that during the coffee fermentation the different groups of microorganisms do not growth exponentially during the process
Zhang, S.J.; De Bruyn, F.; Pothakos, V.; Torres, J.; Falconi, C.; Moccand, C.; Weckx, S.; De Vuyst, L. Following Coffee Production from Cherries to Cup: Microbiological and Metabolomic Analysis of Wet Processing of Coffea Arabica. Appl Environ Microbiol 2019, 85, doi:10.1128/AEM.02635-18
Evangelista et al 2015 Evangelista, S.R.; Miguel, M.G. da C.P.; Silva, C.F.; Pinheiro, A.C.M.; Schwan, R.F. Microbiological Diversity Associated with the Spontaneous Wet Method of Coffee Fermentation. International Journal of Food Microbiology 2015, 210, 102–112, doi:10.1016/j.ijfoodmicro.2015.06.008.
In the three coffee varieties, nine hours after starting the open natural fermentation process, a reduction in the microbial count was evident in some groups of bacteria, in response to the adaptation of the endophytic microbial community in the fermentation process of the total biomass of coffee due to the nutrient conditions, pH and temperature. A reduction in the count of mesophiles, coliforms and yeast was recorded after 18 hours of fermentation. We did not count Acetic Acid bacteria and the count could increase.
We only counted the number of CFU at Zero time. at 9h and 18h during the fermentation process, taking the sample from the fermentation container, at these 3 specific times. But we do not allow the microorganisms to growth in a culture media for 9 or 18 h. At this specific time temperature and pH were different. We did not perform a growing curve or log curve. To explain better: The coffee fruits were pulped and incorporated into the containers to be fermented. At 0h, after 9 and after 18h a sample from 1.2 kg of pulped coffee was collected and processed in the BECOLSUB experimental mucilage remover equipment and the sample of the pulp was collected and processed to count the CFU from each one of the types of microorganism evaluated.
We believed that the increase in total acid as well as lactic and acetic acid in the medium produced by the BAL and BAA bacteria present in the fermentation process, together with the changes in temperature during the 18 hours of the process can generates an adverse condition in the fermentation medium, for example affecting the development of mesophiles and the majority of yeasts. In addition, there is a depletion of nutrients, pH and temperature of the process. The high acidity can favor the growth of mycelial fungi.
- Reviewer
-But still there is no indication how dots on NMDS plot correspond to time points. Please add this info on the Figure 1.
- Authors Answer
We change the figure 1. NMDS analysis of A. Bacteria and B. Fungi associated with the three varieties evaluated and, we add the time points to the figure 1. The change is in Line 512-513.
- Reviewer 1
How authors explain that they isolated strains of Latilactobacillus and Lactococcus but these genera were not detected using sequencing of V3V4-regions of 16S rRNA gene?
-Please check it. And if Lactilactobacillus and Lactococcus classified as “Other”, specify it in Results and Discussion sections.
-Moreover, relative abundance for key taxa (bacterial families level, bacterial genera level, fungi families level, fungi genera level) should be indicated within the text.
- Authors Answer
Latilactobacillus, Lactilactobacillus Lactobacillus and Lactiplantibacillus belong to orden Lactobacillales, Familia lactobacillaceae and Lactococcus to the Familia Streptococcaceae although neither of this 2 families appear in the figure 2 and 3. There is an assignment identified as OTHER in these two figures, in this, the family lactobacillaceae is present as was observed in the 16S.trim.contigs.good.unique.good.fileter.uniqueprecluster.pick.nr_v132.wang.tax.summary.
In addition, we could isolate and identified into pure culture in Var. Colombia different genus (Lactobacillus, Lactococcus and Lactiplantibacillus) from this family
Additionally, we performed the Taxonomic assignment of bacteria in the varieties analyzed at the family level, but including the Top 11 families in abundancy and, this is the figure that we got, in which the family Lactobacillacea is there. Please see the PDF document.
This corroborate that the family was part of assigmnet OTHERS in the figure 2 that is in the manuscript.
We follow your advice and we clarified the information about the Lactobacillaceae family in the abstract line 23.
We add the information in the results from Figure 2, 3, 4 y 5 indicating that only the top 10 taxonomic assignments are shown lines 515, 529 ,586, 601. 523-526. 543-545
We add the information in the discussion in line 669, 681-682,738-740 and 742.
We delete Latilactobacillus because it was not isolated and we change for Lactobacillus that was indeed isolated into pure culture. The changes were done in line 446 and 857
- Reviewer
Line 879- The native microbial communities correspond to microorganisms present in the environment that can be closely related to specific coffee varieties and the place where the coffee is growth that develop during fermentation, which in turn is not a single microorganism but a microbial consortium"
-I know what “native microbial community” means. The question to the authors was about specific key components of communities studied in the work, because coffee microbiomes composition is main focus of the study, the some summary about it should be added to Conclusions section.
- Authors Answer
Following the recommendation of the reviewer, we add the information about what we believe is the composition of this coffee fermentation consortium.
The information was added in conclusion in lines 875-878:
The results indicated that native microbial communities occur naturally during coffee processing. Fermentation depends on a microbial consortium of mixed-acid bacteria (Enterobacteriaceae, Tatumella sp.), lactic acid bacteria (Leuconostoc sp., Weissella sp and Lactobacillaceae.), acetic acid bacteria (Gluconobacter sp and Acetobacter sp.) and fungi Saccharomycodaceae, and Pichia rather than on a single microorganism.

Reviewer 3 Report
Dear Authors,
I think the work is ready to be accepted.
Best regards,
Reviewer
Author Response
Dear Reviewer,
We want to thanks Referee 3 for reviewing the re-submission of the paper. We are very glad that it was possible to answer to satisfaction your inquiries.
Thanks for accepting the manuscript.